**communications** engineering

# An autonomous drone swarm for detecting and tracking anomalies among dense vegetation
Rakesh John Amala Arokia Nathan [1], Sigrid Strand [2], Daniel Mehrwald[1], Dmitriy Shutin [2] &
Oliver Bimber [1] ✉

Swarms of drones offer increased sensing aperture. When these swarms mimic natural behaviors, sampling is enhanced by adapting the aperture to local conditions. We demonstrate that this enables detection and tracking of heavily occluded targets. Object classification in conventional aerial images generalizes poorly due to occlusion randomness and is inefficient even under minimal occlusion. In contrast, anomaly detection applied to synthetic-aperture integral images remains robust in dense vegetation and independent of pre-trained classes. Our autonomous, centralized swarm searches for unknown or unexpected occurrences, tracking them while continuously adapting its sampling pattern to optimize local viewing conditions. We achieved average positional accuracies of 0.39 m with average precisions of 93.2% and average recalls of 95.9%. Here, adapted particle swarm optimization considers detection confidences and predicted target appearance. We present a new confidence metric that identifies the most abnormal targets and show that sensor noise can be effectively included in the synthetic aperture process, removing the need for costly optimization of high-dimensional parameter spaces. Finally, we provide a hardware-software framework enabling low-latency transmission and fast processing of video and telemetry data. Although our field experiments involved six drones, ongoing technological advances will soon enable larger, faster swarms for military and civil applications.

In recent decades, the use of unmanned aerial vehicles (UAVs), or simply drones, equipped with a range of sensors for environmental perception has skyrocketed. There is a plethora of UAV types and sizes, ranging from inexpensive toy-like kits for DIY assembly to professional counterparts, including VTOL (Vertical Take-Off and Landing) and more capable MALE and HALE (Medium-/High-Altitude Long-Endurance) drones. Their applications are similarly diverse. It is now difficult to imagine use cases in environmental monitoring, industrial inspection, search and rescue, and numerous security applications that do not rely on aerial information somehow delivered by the drones. Once the stuff of science fiction, approaches involving multiple cooperating drones – referred to as drone swarms – have in recent years become engineering concepts and prototypes.

Unlike single-drone systems, drone swarms can perform complex tasks by utilizing decentralized operation and data processing, and can thus exhibit behavior which is similar to – or even mimics – that observed in natural swarms of, for instance, some insect[1], fish[2], and bird[3] species. The key advantages of such approaches are, among other things, robustness of the swarm to external and internal perturbations and disruptions, and efficiency in data collection. Most importantly, however, swarms offer an increased spatio-temporal sensing aperture - a conceptually different approach to perception that can otherwise not be easily realized with a single-drone system; in other words, swarms offer an advantage when capturing spatial phenomena that change with time. In fact, for stationary or time-invariant phenomena (or those that can be assumed to be so), a single drone can be utilized to collect spatial data[4–6]. Clearly, robustness and efficiency aspects specific to swarms are not the focus of the single-drone approaches mentioned above; but when time aspects, specifically, time-dependencies of the phenomena observed, play a role, swarms offer capabilities that are otherwise inaccessible to single-drone systems. Examples, here not limited to UAVs, include monitoring rapidly evolving ocean phenomena[7], navigation and synchronization in GNSS-denied environments[8], and target detection and tracking[9]. In our work, we concentrate primarily on the use of drone swarms specifically for aerial sensing of temporal phenomena by exploiting the increased sensing aperture that the swarms offer.

[1]Department of Computer Science, Johannes Kepler University Linz, Linz, Austria. [2]Institute of Communications and Navigation Communications Systems, German Aerospace Center, Oberpfaffenhofen, Wessling, Germany. ✉e-mail: oliver.bimber@jku.at

**Fig. 1 | A swarm of drones collectively samples the optical signal of an extremely wide adaptable airborne lens.** Each individual drone recording (bottom) covers a large depth of field that is given by the narrow physical aperture of the integrated camera lenses. Here, heavily occluded targets are only fractionally visible. By integrating these images, we mimic the extremely shallow depth of field of a very wide synthetic aperture lens (size equals the sampling area of the swarm). Out-of-focus occlusion is thus suppressed, while targets in the synthetic focal plane (e.g., on the forest floor) are emphasized (right). Since the sampling formation of the swarm can be dynamically controlled based on logical viewing conditions (e.g., density of the forest), the synthetic aperture is adaptable and not fixed.

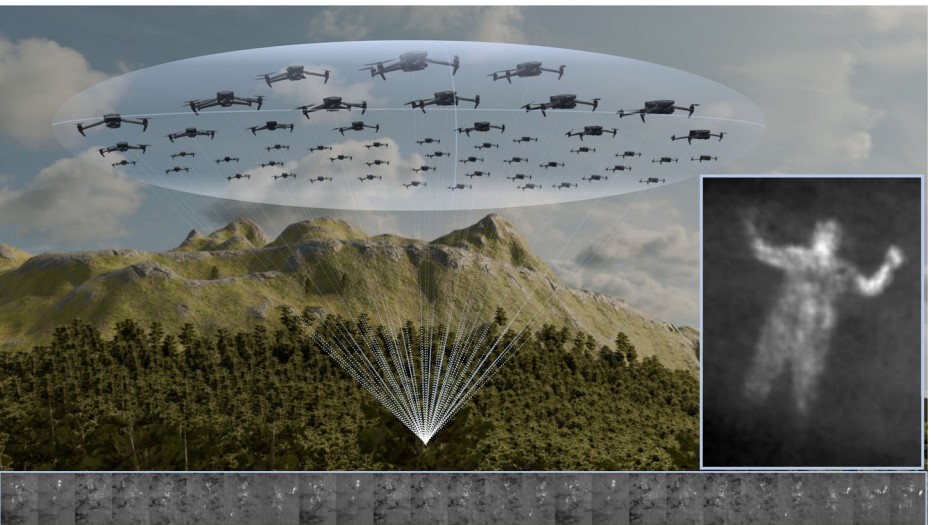

Aerial sensing with drone swarms has been explored in a number of studies[10,11]. One study[12] specifically addressed challenges of swarm deployment for general surveillance applications. Agricultural applications of swarms for aerial inspections and monitoring have also been investigated[13]. A particularly relevant use case of aerial imaging in forested areas emphasizes the importance of a swarm approach for forest survival and exploration[14]. Furthermore, a swarm of micro-UAVs capable of navigating through dense forests while performing mapping and solving navigation tasks has been developed and demonstrated[15], highlighting challenging scenarios with a number of important use cases in both civil and military application areas.

A common problem of conventional aerial image sensing with drones is occlusion caused by vegetation, such as forests, which usually makes it impossible to find, detect, and track targets such as people, animals and vehicles in aerial videos. Several state-of-the-art drone swarm approaches[16,17] rely on merging machine-learning based image classification results of single drone views in the hope that at least one view has little or no occlusion and thus provides confident classification outcomes. This, however, fails in the case of denser occlusion, as no view achieves this level of confidence. In fact, it is the randomness of occlusion that prevents a neural network from successfully generalizing occluded cases[18,19]. Using the Airborne Optical Sectioning (AOS) imaging method[20], we solve this problem with a special scanning principle known as synthetic aperture (SA) sensing[18–28]. Today, synthetic aperture sensing is used in many fields, such as radar[29–31], interferometric microscopy[32], sonar[33], ultrasound[34,35], LiDAR[36,37], imaging[38,39], and radio telescopes[40,41]. Similar to interlinking distributed radio telescopes to improve measurement signals by coherently combining individual receptions, AOS integrates optical images recorded over a large forested area in order to computationally remove in real time occlusion caused by trees. This creates extremely shallow depth-of-field integral images with a largely unobstructed view of the forest floor. The unique advantages of AOS, such as its real-time processing capability and wavelength independence, open up many new applications in contexts where occlusion is problematic. These include, for instance, search and rescue[18,19,21,27], wildlife observation[22], archeology[20], wildfire detection[23], and surveillance. Another advantage of AOS is that it permits integration of processed images rather than raw camera frames. Thus, anomalies (i.e., pixels that, compared to surrounding pixels, are abnormal in terms of, e.g., color, temperature, and optical flow) can be pre-identified with modern anomaly detectors[42–45] and integrated for occlusion removal[24]. In cases of dense occlusion, integrating anomalies, as well as detecting and tracking occlusion-suppressed clusters of anomalies, is much more reliable than first classifying partially occluded objects in regular aerial images and then integrating classification results for detection and tracking[27]. Unlike

classification, anomaly detection requires no training data and is not limited to a predefined set of classes, but it also does not support differentiation between classes. Since AOS combines video frames that are captured blindly (i.e., without considering local viewing conditions, such as forest density) by a single drone during flight, it has to date remained difficult to detect and track occluded moving targets efficiently[25,26].

Here, we present a dynamic approach to airborne synthetic aperture imaging that explores and considers local sampling conditions. If we take a classical fixed sampling pattern in the SA plane (i.e., a constant set of waypoints above forest at which a single drone captures images sequentially) as an extremely wide but static airborne lens with constant optical properties, then a swarm of drones effectively collects an optical signal of extremely wide adaptable airborne lenses in parallel (see Fig. 1). Note, that this collection requires a centralized architecture. In our previous work[27], we demonstrated in simulations that autonomous drone swarms significantly outperform blind sampling strategies of single drones and are therefore efficient in detecting and tracking occlusion-suppressed anomalies of partially occluded targets. Here, exploration and optimization of local viewing conditions (such as occlusion density and target view obliqueness) using an adapted particle swarm optimization (PSO) provided much faster and much more reliable results. See Supplementary Movie 1 *(Introduction)*.

In this article, we report on three main contributions. First, we discuss a hardware implementation of our fully autonomous drone swarm strategy and present the results of real-life field experiments with it. Among other things, this required the development of a hard- and software infrastructure for 70–120 Mbit/s video and telemetry transmission and processing, and for synchronous swarm control. Without knowledge of the target to be detected and tracked, our swarm explores general and possibly partially occluded anomalies on the ground (i.e., portions of abnormal colors or temperatures). It strives to continuously optimize the target's visibility by efficient and dynamic synthetic aperture sampling using a modified PSO. The criticality of precise image registration opposes sensor errors (in particular, rotational heading drifts) of multiple collaborating drones. Therefore, our second contribution is to demonstrate that sensor noise can effectively be included in the SA image integration process, rendering a computationally costly optimization of high-dimensional parameter spaces unnecessary. Third, we enhanced our PSO and objective function to consider detection confidences and predicted target appearance.

In *Results*, we first outline our SA and sensor noise integration principle as well as our PSO strategy and objective function. We then present the results of three field experiments with the swarm: I. A tracking and classification task in which the swarm detected, classified, and tracked the anomaly of a target (color signal of a moving car) in an open field and sparse

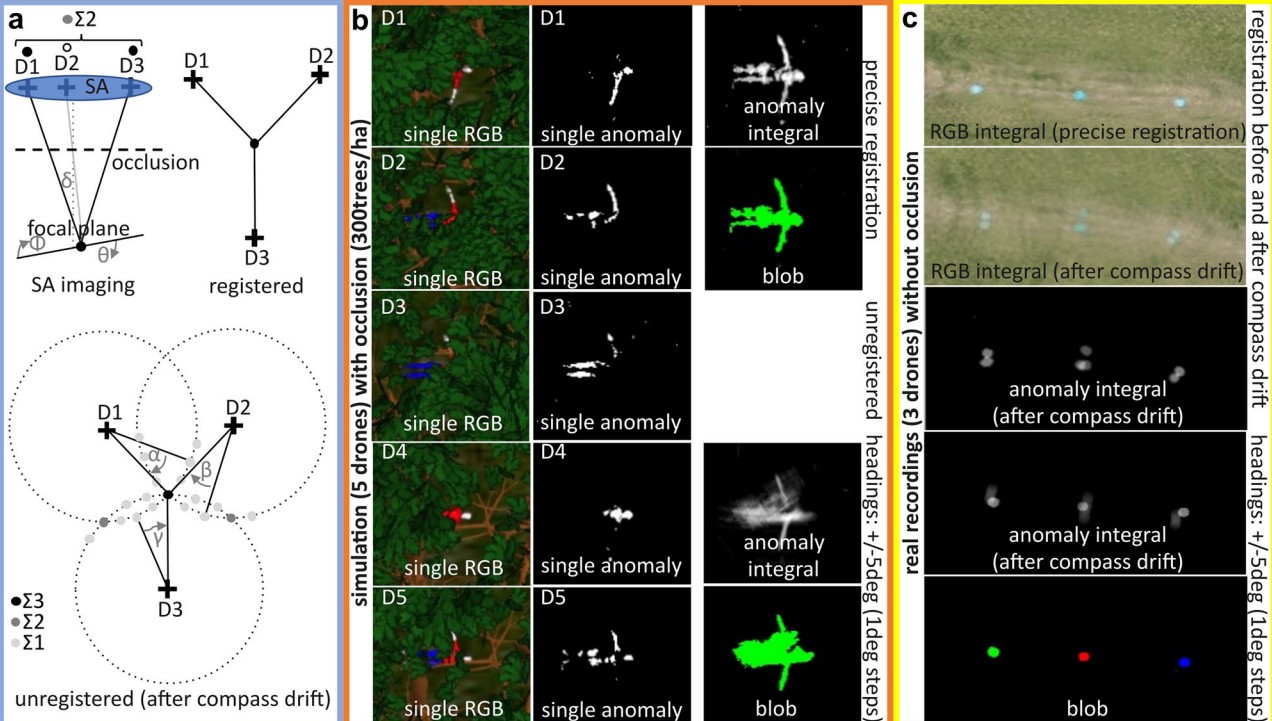

**Fig. 2 | Integration Principle. a** Integration principle illustrated with three drones (D1, D2, D3) for occlusion suppression in a defined synthetic focal plane (SA imaging) and for the suppression of registration artifacts due to unknown heading parameters (registered vs. unregistered). **b** Simulation example for 5 drones (D1-D5) and an occluded person lying in a forest (300 trees/ha). Single RGB images (left column). Corresponding single anomaly images (center column). Occlusion-suppressed anomaly integrals and blob detection results for the cases of precise registration and for the case of imprecise registration (due to heading errors) after integrating 10 heading variations in 1° steps (right column). Details of the simulator are provided[27]. **c** Real recordings by three drones of three unoccluded circular targets on the ground. Top, middle, bottom: RGB integrals after precise initial registration and after misregistration due to compass drift. Anomaly integral after compass drift. Anomaly integral including heading integration after compass drift and final blob detection.

forest with no or very little occlusion. II. A search task in which the swarm was interactively deployed to regions of interest before exploring them autonomously to search for anomalies of potential targets (temperature signal of lying and standing/walking persons) and to detect them in a dense forest with heavy occlusion. Finally, III. a tracking task in which the swarm autonomously detected and tracked moving anomalies (temperature signal of walking persons) in a dense forest with severe occlusion. In our experiments, we achieved an average positional accuracy of 0.39 m with an average precision of 93.2% and an average recall of 95.9%. We summarize and discuss our results and findings in *Discussion*, while implementation and evaluation details are provided in *Materials and Methods* and in the Supplementary Materials *sections*. To our knowledge, this is the first fully autonomous drone swarm implementation that is able to detect and track general targets that appear abnormal within their environment under dense and realistic occlusion conditions.

## Results
### Integrating occlusion and sampling parameters
The basic principle of synthetic aperture imaging is the integration of multiple images with individually weak visual cues into one integral image with an improved signal. To remove occlusion, which might dominate in conventional aerial recordings of a forest, integration of multiple images captured by drones in various poses within a larger sampling area suppresses occlusion. As illustrated in Fig. 2A (SA imaging), the sampling area corresponds to the SA area, and for integration ($\Sigma$) single drone recordings (D1, D2, D3,...) must be registered and averaged with respect to a defined synthetic focal plane at a distance $\delta$ and with orientation ($\theta, \Phi$) relative to the SA. See Supplementary Note 1 (cf. Supplementary Fig. S1) of the Supplementary Materials for implementation details. A point in the focal plane might be occluded in some drone recordings, while it is visible in others. It has been

shown that, when a uniform occlusion volume (uniform occluder shapes and spatial distribution) is considered, the degree of visibility is inversely proportional to the degree of occlusion for conventional images, but that this relationship does not hold for integrated images[28]. Since occlusion volumes in typical forests are rarely uniform, consisting rather of regions with varying densities, a sampling strategy that adapts itself to these local differences relaxes the occlusion-visibility relationship even more[27]. As shown in Fig. 2B, applying an anomaly detector, such as a Reed-Xiaoli Detector (RX) ([42,43], see Supplementary Note 2 of the Supplementary Materials for implementation details), to the drone images before integration divides pixels into two sets: Those that appear abnormal with respect to their surroundings, and those that do not. Integrating these binary anomaly masks instead of the raw camera images results in occlusion-suppressed anomaly clusters[24] that can be detected and tracked (with, e.g., blob detection and tracking[46,47]). Here, pixel values of such anomaly integrals approximate visibility. For instance, a pixel with a value of 3 in an anomaly integral computed from 5 drone images indicates that the scene point corresponding to this pixel has been seen by three out of the five drones, and hence the resulting visibility is 60%.

The integration process explained above requires precise information of drone poses and the underlying terrain. However, drone sensors used for pose estimation, such as GPS, IMU, and digital compass, have limited accuracy and precision. Digital compass modules used to derive a drone's heading drift are particularly prone to severe drifting even in short time periods. While positional errors of real-time kinematics GPS (RTK) are in the centimeter range and tolerable, rotational errors of a few degrees only (e.g., caused by imprecise heading estimations) readily lead to total misregistrations and failure of image interpretation. Thus, assuming an appropriate camera-gimbal stabilization and RTK precision, it is mainly the following integration parameters that remain unknown: (1) the exact

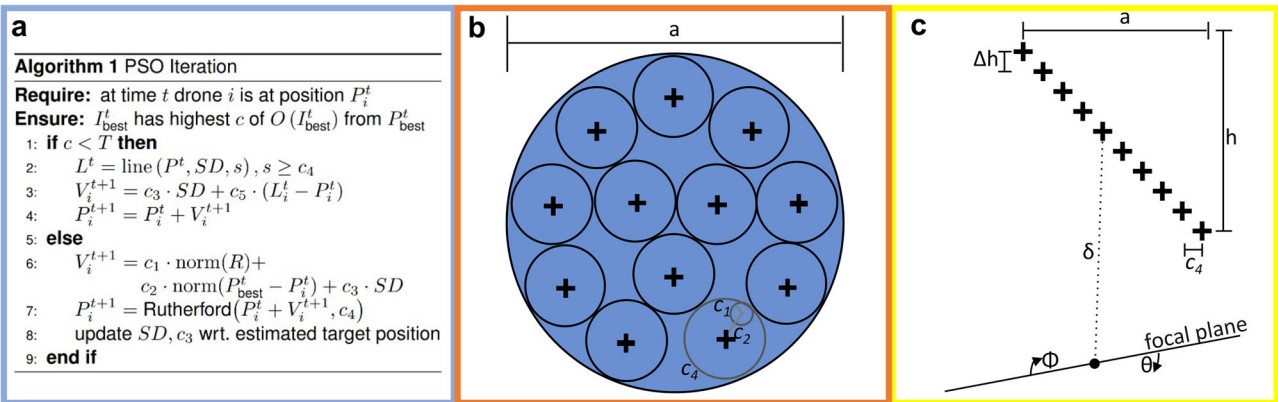

**Fig. 3 | Swarm Formation Principle. a** Pseudocode of PSO iteration with divergence branch (lines 2-4) and convergence branch (lines 6-8). **b** The *packing-circles-in-circle solution* of our minimal horizontal distance constraint defines the relationship between PSO hyperparameters and SA diameter $a$. The smaller circles indicate the exploration+exploitation action radius of each drone during one PSO iteration: $c_1 + c_2 \leq c_4$ (e.g., see gray circle). **c** A relative altitude difference $\Delta h$ between drones to avoid collision and self-occlusion.

synthetic focal plane parameters $(\delta, \theta, \Phi)$, that is, the distance and orientation of the observed ground surface, which can only be estimated (e.g., from the drones' height above ground level as measured by GPS or pressure sensors and optionally a digital elevation model of the surrounding terrain), and (2) the individual headings of each drone in the swarm. With $N$ drones in the swarm, we are looking at an $(N + 3)$-dimensional parameter space to be explored for an optimum. Brute-force searching this parameter space in $n$ equal steps for each parameter would naturally require a total of $n^{N+3}$ steps. Considering only $N = 5$ drones and $n = 10$ steps would require 100,000,000 optimization steps. Even if a more efficient optimization strategy could reduce this number, performance would be far from real time. In fact, each step requires image registration, averaging, and evaluation of the resulting integral image by an objective function - a very time-consuming process. The approach we propose to overcome this challenge is not to search for an optimal parameter solution, but to include all considered options directly in the integral images. For the case of unknown heading values of three drones only, this technique is shown in Fig. 2A. With precise heading values, a common point that appears in multiple drone images at different locations is perfectly registered in the integral image, as shown in Fig. 2A (registered). In the case of compass drift, this is no longer the case. As illustrated in Fig. 2A (unregistered), a common point now also appears at multiple different places in the integral image (relative to the unknown heading offsets – here, $\alpha$, $\beta$, and $\gamma$). Now, consider integrating multiple instances of the binary anomaly image of each drone, with each image being transformed (i.e., rotated in the case of headings) independently in $n$ steps within a certain heading range. We can then expect that the visibility values in the resulting anomaly integral are maximized at these common locations when images are transformed "correctly", that is, with an optimal parameter set $(\alpha, \beta, \gamma)$, as more drones now contribute to these locations. Integration values that result from incorrect parameter sets for other locations remain present in the integral image, but are suppressed as fewer drones contribute there. In Fig. 2B, this is shown for $n = 10$ heading steps (each 1°) within a $+/-5°$ range. While Fig. 2B shows simulations with occlusion, Fig. 2C presents real recordings without occlusion. For both examples, the anomaly blobs detected appear at the same positions (although in slightly different shapes) in the registered and in the unregistered case. While the parameter optimization example from above would require $n^{N+3} = 100,000,000$ steps, our parameter integration would now require only $Nn + 3n = (N + 3)$ $n = 80$ steps. This is easy to achieve in real time.

Therefore, we integrate $(N + 3)n$ images per PSO iteration (i.e., the sampling constellation of the swarm determined by the PSO at one particular instance in time). This suppresses occlusion while simultaneously considering unknown sampling parameters. Below, we explain how these sampling constellations are determined.

## Swarm formation and sampling

Our objective is to detect and track a single target that appears most abnormal and, if possible, to enhance its visibility over time. For an anomaly integral $I^t$ that is sampled from a particular swarm constellation at time $t$, we first filter out noisy pixels of low visibility (i.e., integrated visible abnormality values below 10-14%) and then determine clusters of connected pixels using blob detection[47]. If geometric feature constraints of the target are known (e.g., minimum/maximum size, or shape constraints), we remove outliers. For each remaining cluster in $I^t$, we then determine its relevance, which is the integral of its anomaly pixel values over the cluster area. Recall that each single anomaly pixel value approximates visibility (visible abnormality) of the corresponding target point, that is, how often the sampling drones of the swarm have seen this point. Thus, the cluster integral approximates the visibility of the entire target, that is, how often the sampling drones of the swarm have seen all of the target's points. Large, well-visualized anomaly clusters are therefore considered more abnormal than those that are small and less visible. The cluster with the maximum relevance is detected and tracked if we are sufficiently confident that it can really be considered an anomaly. In our case, confidence is determined as the relevance ratio of the two clusters with the greatest and the second-greatest relevance. If this confidence is one (or near one), then at least two similar clusters exist, and our single-target anomaly objective fails. Thus, our objective function $[b,c]$ $=O(I^t)$ takes an anomaly integral as input and returns the most relevant cluster blob $b$ and the confidence $c$ that was determined for it. Note that the legacy confidence metric[27] identified the most abnormal target based on the maximal contour size of connected pixel clusters in the integrated anomaly masks. This approach equated the largest spatial anomaly in the integral image with the highest abnormality—a fundamental limitation, as contour size reflects only spatial extent, not true anomaly severity. Our proposed metric addresses this by integrating the anomaly values over the target area, thereby jointly considering spatial scale and anomaly intensity. Consequently, smaller targets exhibiting stronger anomalies can now be prioritized over larger ones.

Algorithm 1 (cf. Fig. 3A) summarizes one iteration of our particle swarm optimization (PSO) to maximize confidence $c$ of $O(I^t)$ by balancing exploration and exploitation strategies. The outcome of each iteration at time $t$ is a new swarm constellation (drone positions) for sampling in the next iteration at time $t + 1$. For all raw images that are sampled by each drone during one iteration, we determine binary anomaly masks and integrate them (including integrating unknown sampling parameters) to obtain $I^t$, as explained earlier. The reference perspective for this integration is computed, but it can be based on any drone position in the swarm. We chose the drone position $P^t_{best}$ with corresponding $I^t_{best}$ for which $c$ is maximal. If $c$ is greater than a confidence threshold $T$, then a target is

**Fig. 4 | Detection, Tracking, and Classification of Moving Targets in Sparse Forest.** Under simple conditions (no or little occlusion), the swarm detects and tracks the target that appears most abnormal (a moving vehicle). **a** A satellite image with the ground-truth path of the vehicle (yellow) and the path of the tracking swarm (swarm's center of gravity, blue line), individual drone positions of the swarm at time $t$ (small light blue circles), best sampling position at time $t$ (small yellow circle), and drone movement between time steps $t-1$ and $t$ (white arrows). **b, c** Close-ups at various times (drones encircled). **d** Visual results of RGB and anomaly integrals, blobs detected, and classification results at various waypoints (indicated by circles in **a** and by matching frame colors in **d**). This experiment was recorded in Supplementary Movie 2 *(Experiment I)*.

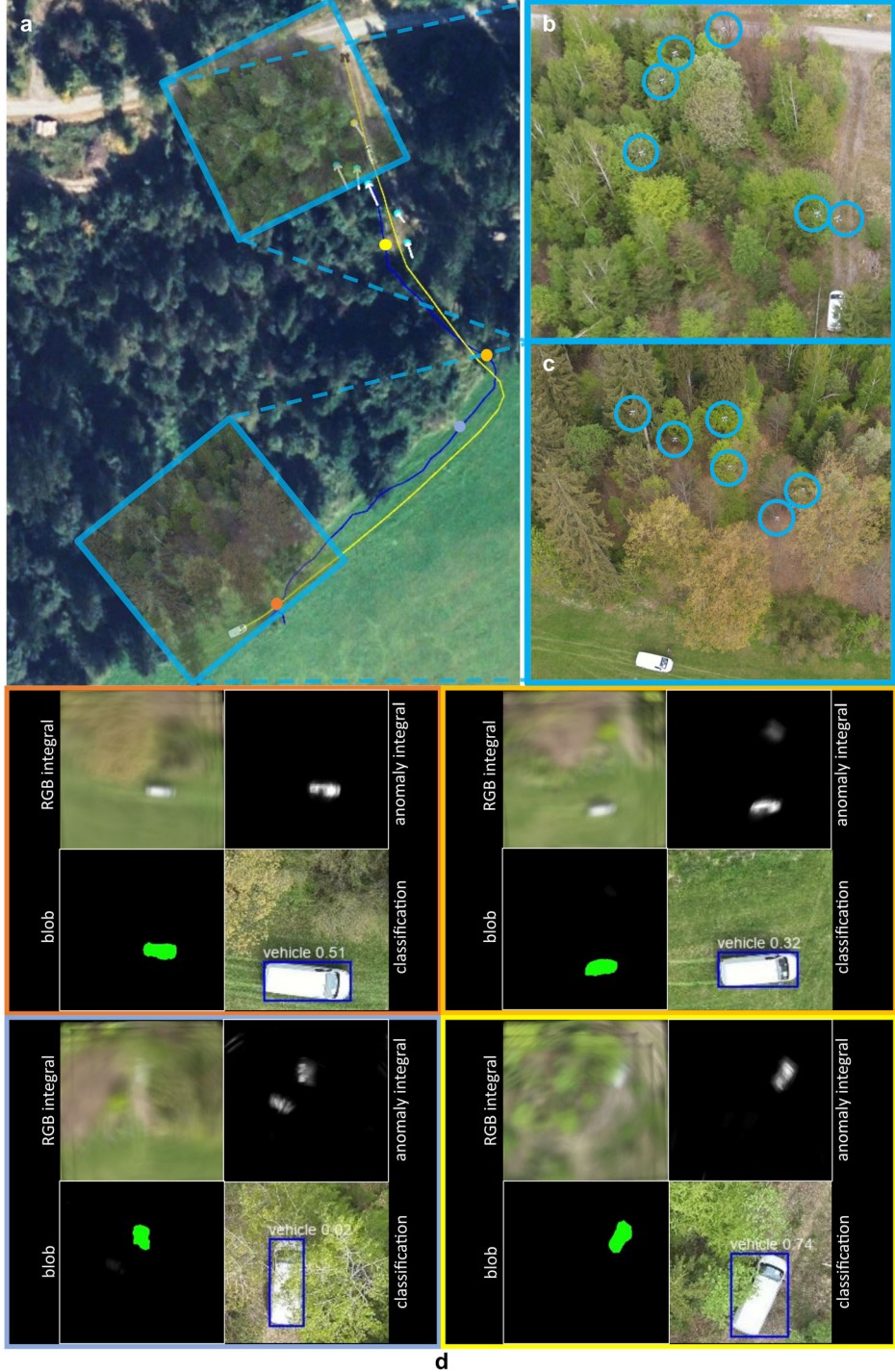

detected and the swarm will converge to track it (lines 6-8). For instance, $T = 2$ (determined empirically to be suitable for all experiments) would require that, for a cluster to be considered the most abnormal, it must appear at least twice as abnormal as the second-most abnormal one. Otherwise, we declare that no target is found or that a previously found target has been lost. In this case, the swarm will diverge towards a scanning direction $SD$ in which a target is most likely to be (lines 2-4). Here, $SD$ is either a predefined heading if no target has ever been found or the direction vector between the swarm center and the position of a previously found but lost target. Line 2 interpolates (over multiple iterations) new swarm constellations towards a linear formation $L^t$ that progresses forward at distance steps $s$ and in a direction orthogonal to $SD$. New velocity vectors $V_i^{t+1}$ and drone positions

$P_i^{t+1}$ for the sampling constellation in the next iteration are then determined in lines 3 and 4 according to hyperparameters $c_3$, $c_4$, and $c_5$. If a target is detected, then the new velocity vectors (line 6) are determined based on (i) an exploration term that scales a normalized random vector $R$ by hyperparameter $c_1$, (ii) an exploitation term that provides a scaled (by hyperparameter $c_2$) bias towards $P_{best}^t$, and, finally, (iii) a term that enforces a bias towards the last observation of the target ($c_3 \cdot SD$). Using Rutherford scattering[48], new drone positions for the next swarm constellation (line 7) are constrained to a minimum horizontal distance $c_4$ between drones.

The hyperparameters used are defined as follows: $c_1$ and $c_2$ represent the degrees of exploration and exploitation, as is common in PSO, $c_3$ is the estimated speed of the target (determined in line 8) or the swarm's initial

scanning speed before a target has been found, $c_4$ is the enforced minimum distance between each drone in the swarm, and $c_5$ is the speed at which drones diverge to form up into a linear scanning formation if a target has either been lost or not been found.

As explained[27] and illustrated in Fig. 3B, Rutherford scattering drives the swarm to an approximated *packing-circles-in-circle solution*[49] with an SA diameter of $a = c_4 \cdot r_N$, where $r_N$ is the packing number[50,51] for $N$ drones in the swarm (e.g., $r_N = 3.813$ for $N = 10$ or $r_N = 3.0$ for $N = 6$). Therefore, the larger $N$ and $c_4$, the larger the swarm's synthetic aperture becomes, which reduces the depth of field in the resulting integral images and image overlap of individual drones in the synthetic focal plane. Ensuring our minimum horizontal distance constraint thus implies $c_1 + c_2 \leq c_4$. To keep exploration from overtaking exploitation, we also require that $c_1 \leq c_2$. Note that $c_1$, $c_2$, $c_3$, and $c_4$ are all in distance units (meters in our case), while $c_5$ is a normalized scalar in the interval [0.1] that controls the smoothness of divergence to a linear formation if the target is lost.

To avoid collisions with other drones in the swarm, each drone is operated at different altitudes, as shown in Fig. 3C. The relative altitude difference $\Delta h = c_4/(N-1)/tan(FOV/2)$ is chosen such that no drone appears in the field of view (*FOV*) of another drone's camera during image sampling[27]. We have shown[27] that the swarm's height differences of $h = (N-1) \cdot \Delta h$ have no significant effect on image integration and on the resulting integral image. We add a safety margin to $\Delta h$ to take into account GPS errors and downwash turbulence during overflights. Results of downwash tests for various $\Delta h$ and flight speeds are presented in Supplementary Note 5 (cf. Supplementary Fig. S4) of the Supplementary Materials. Note that the absolute height of the swarm is set based on environmental conditions, such as tree heights. Since sampling at lower altitudes requires a smaller distance to be covered for occlusion removal[20,28], we prefer to operate the swarm at a minimal distance above the tree canopy.

Below, we discuss our choice of parameters with the individual experiments. The average processing time (image registration and integration, anomaly and blob detection) per PSO iteration for a swarm of six drones was approximately. 600 ms for all experiments. Approximately 80 ms round-trip time had to be added for uploading waypoint data and downloading video and telemetry data, and for the flight time from one formation to the next (which depends on waypoint distances and flight speed).

## Experiment I: Detection, tracking, and classification of moving targets in sparse forest

In our first field experiment, we investigated a relatively simple detection and tracking case with no or little occlusion. A moving vehicle was expected to be the target that appeared to be the most abnormal amidst a mix of sparse forest and meadow ground (cf. Fig. 4A). The swarm consisted of 6 DJI Mavic 3 T drones equipped with Real Time Kinematic (RTK) modules for precise positioning. The drones were operating autonomously at a height of 45-55 m above ground level (AGL), as explained above (cf. Fig. 4B, C). Color (RGB) images were captured and processed. The selected hyperparameters were $c_1 = 1$ m (exploration), $c_2 = 2$ m (exploitation), $c_3 = 2$ m (initially estimated target speed), $c_4 = 4.2$ m (minimum horizontal drone distance with safety margin), $c_5 = 0.3$ (speed coefficient for divergence), $s = c_4 = 4.2$ m (horizontal distance in linear scanning configuration), $SD = 316°$ (initial scanning direction), $\Delta h = 2$ m (minimum vertical distance with a FOV = 43° and safety margin), $T = 2$ (confidence threshold) - which resulted in an SA with a minimum diameter of $a = 12.2$ m and a height of $h = 10$ m (cf. Fig. 3B, C). The drones' flight speed was 3 m/s. A mobile networked RTK sensor (see Supplementary Note 4 of the Supplementary Materials) was attached to the vehicle and also provided ground-truth positioning.

Figure 4A visualizes the ground-truth path taken by the vehicle (at a maximum speed of approx. 5 km/h) and the path flown by the swarm (the swarm's center of gravity was used as a reference) to track it. Figure 4D shows visual results of RGB integrals, anomaly integrals, and blobs detected at various positions along the tracking path (indicated in Fig. 4A). In addition to our approach, we apply a simplified classifier for object

recognition (a pretrained YOLO-World v2 model[52] was employed, retained only the 'vehicle' and 'person' classes – focusing exclusively on detecting and tracking these targets in this hypothetical use case) to demonstrate limited effectiveness of object classification in complex forested terrain with partial occlusion[18]. Object recognition was carried out on the image captured from the drone perspective corresponding to the best sampling position $P_{best}^t$, i.e., the viewpoint for which the computed integral $I_{best}^t$, representing our confidence measure c, is maximal. This viewpoint also yielded the highest YOLO confidence score among all images captured by the swarm at time t. Recognition results and confidence scores are shown in Fig. 4D. In the Supplementary Note 3, Table S1 lists and Fig. S2 plots all relevant quantitative measures for all PSO iterations of this experiment. In summary, we achieved a detection precision of 93.87% and a recall of 100%, while the error between ground-truth target position and position estimated by the swarm was on average 0.26 m. Our own confidence metric delivered high scores (on average $c = 147.1$, and always far higher than our confidence threshold of $T = 2$). However, we obtained relatively low YOLO confidence scores (0.4 on average and as low as 0.02 for partial occlusion) of the pre-trained classifier - even for such simple occlusion conditions. Only by choosing a low YOLO confidence threshold (of 0.01 in this case) and restricting the classifier to only two classes could an adequate classification result be achieved. Although better classification results might be achieved by classifiers that are specifically trained for a particular task (e.g., on aerial images of ground targets), we argue that a higher degree of occlusion would render classification unfeasible, as the inherent randomness of occlusion is not well generalizable[18]. This experiment was recorded in Supplementary Movie 2 *(Experiment I)*. Re-evaluating against the legacy method[27] in an open-loop manner, using only the images captured by the drones as done with YOLO for automatic classification (ignoring sensor noise and using its original confidence metric) shows a precision drop from 93.9% to 91.8%. This occurs under an optimal threshold configuration yielding 100% recall. Note that precision would be further degraded by sensor noise exceeding the levels observed during the experiment.

## Experiment II: Detection of localized targets in dense forest and swarm deployment

In the next experiment, we explored much denser occlusion conditions in which conventional classification failed entirely. Here, the task was to detect (rather than track) localized people who appear abnormal in terms of temperature within a dense forest when using thermal imaging. As in Experiment I, the swarm consisted of 6 DJI Mavic 3 T drones with networked RTK. We utilized a 7th (deployment) drone (a DJI Matrice 30 T flying 23 m above the swarm) to initially deploy the swarm quickly to a designated exploration area before releasing it to explore the area autonomously. This deployment strategy is implemented as follows: As long as the deployment drone was moving, the swarm followed it by constantly matching the swarm's horizontal center of gravity with the deployment drone's horizontal position. Once the deployment drone stopped, the swarm autonomously explored the area based on Algorithm 1 (Fig. 3A). It should be noted that swarm deployment serves only to guide the swarm initially to a region of interest and does not influence the autonomous exploration strategy. The hyperparameters for the autonomous exploration used in this experiment were $c_1 = 1.7$ m, $c_2 = 3.42$ m, $c_3 = 3$ m, $c_4 = 5.15$ m, $c_5 = 0.3$, $s = c_4 = 5.15$ m, $SD = 302°$, $\Delta h = 2$ m (FOV = 35°), $T = 2$ (confidence threshold) - which resulted in an SA with a minimum diameter of $a = 15.45$ m and a height of $h = 10$ m. Flight heights were 45-55 m AGL, and flight speed was 2 m/s.

In this experiment, we aimed to find one lying and one standing person – both heavily occluded – in two different densely forested areas under low light conditions (a few minutes before sunrise). The targets were located in distinct deployment regions, such that at each deployment location, only one target fell within the swarm's sensing footprint, while the other target remained outside the sensing range. Note that, while thermal images were recorded and processed by the swarm for detection processing, RGB images were captured solely by the deployment drone for observation

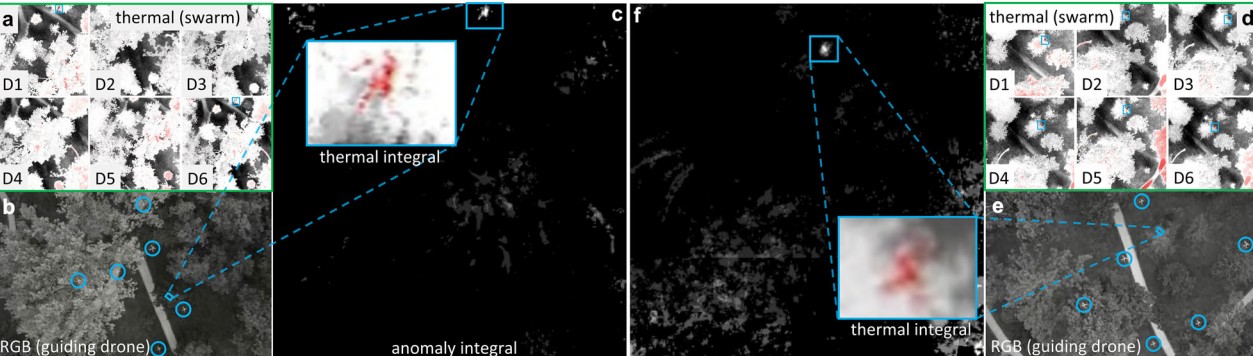

**Fig. 5 | Detection of localized targets in dense forest and swarm deployment.** To find lying (**a**–**c**) and standing (**d**–**f**) persons in a dense forest under low light conditions, the swarm was deployed to search areas in which the targets were expected to be, and then it explored these areas autonomously. **b**, **e** The view (RGB) from the deployment drone, with swarm drones encircled. **a**, **d** The views of the swarm drones (D1…D6, target positions indicated by boxes). **c**, **f** Thermal and anomaly integrals also reveal the person anomalies, which remain hidden in the single images. Note that thermal images are color-coded (hot = red, cool = gray). This experiment was recorded in Supplementary Movie 3 *(Experiment II)*.

purposes only. Visual results for both cases are shown in Fig. 5. Quantitative detection results are shown in Table S2, and Table S3 compares detection results of our approach with the legacy method in the Supplementary Note 3. In contrast to tracking moving targets (see Experiment I and III) detecting stationary targets requires only one correct detection after deployment. This is the case for our new approach in this experiment, while the legacy method fails. The errors between the targets' ground-truth bounding boxes and positions estimated by the swarm were 0.33 m for the lying person and 0.08 m for the standing person. Note that, after successful detection, a reliable automatic classification (e.g. using a classifier such as YOLO) under such extreme occlusion conditions is not realistic (see Experiment I), manual classification by a human operator by visually observing the thermal and anomaly integrals computed might be more efficient (cf. Fig. 5C, F). This experiment was recorded in Supplementary Movie 3 *(Experiment II)*. None of the targets (neither lying nor standing person) was detected with the legacy method[27] (ignoring sensor noise and using its original confidence metric) shows that.

### Experiment III: Detection and tracking of moving targets in dense forest

The objective of this experiment was to use the swarm to autonomously detect and track walking persons that appear most abnormal in terms of temperature through dense forest. As in the previous experiments, the swarm consisted of 6 thermal-imaging DJI Mavic 3 T drones equipped with RTK. The personnel on the ground were carrying a mobile RTK-equipped navigation device for ground-truth measurements. The hyperparameters used in this experiment were identical to those used in Experiment II. The targets to be detected and tracked were two persons walking side by side at a maximum speed of approximately. 3 km/h. We chose two persons walking side by side as targets in this experiment to extend our use cases over single-person targets (Experiment II) and vehicles (Experiment I).

Figure 6A visualizes the ground-truth path the persons walked and the path the swarm flew (swarm's center of gravity was used as a reference) to track them (close-ups are shown in Fig. 6B, C). Figure 6D presents visual results of thermal integrals, anomaly integrals, and blobs detected at various positions along the tracking path (indicated in Fig. 6A). In the Supplementary Note 3, Table S4 lists and Figure S3 plots all relevant quantitative measures for all PSO iterations of this experiment. In summary, we achieved a detection precision of 92.3% and a recall of 92.3%, while the error between ground-truth target position and position estimated by the swarm was on average 0.53 m. Our own confidence metric delivered high scores (on average $c = 27.6$, with $c < T$ in only 4 out of 56 PSO iterations where the target was lost but immediately re-detected in the next iteration). While false negative detections result from temporary invisibility (e.g., due to locally

dense vegetation), false positive detections occur when extraneous features appear more anomalous than the actual target, such as larger hotspots on the ground. This experiment was recorded in Supplementary Movie 4 *(Experiment III)*. Re-evaluating against the legacy method[27] in an open-loop manner, using only the images captured by the drones as done with YOLO for automatic classification in Experiment I (ignoring sensor noise and using its original confidence metric) shows a precision drop from 92.3% to 89.3%. This occurs under an optimal threshold configuration yielding 100% recall. Note that precision would be further degraded by sensor noise exceeding the levels observed during the experiment.

### Discussion

Swarms of drones offer (i) an increased sensing aperture and (ii) enhanced sampling because imitating behaviors observed in natural swarms allows them to adapt the aperture to local conditions. Here, we have shown that detecting and tracking heavily occluded targets is feasible with such an approach. Anomaly detection applied to integral images is robust under heavily occluded conditions and independent of pre-trained classes. Object classification applied to conventional aerial images, however, is inefficient even under lightly occluded conditions because it cannot generalize well the randomness of occlusion.

Anomaly detection allows the environment to be explored for things that are unknown or unexpected, but does not provide class information about them. Hence, it is possible that pronounced anomalies that are not the target are tracked and detected. A promising avenue for future research is investigating synergies between object classification and anomaly detection, along with developing deep-learning anomaly detectors adaptable to specific environmental conditions. Applying zero-shot classification[53] to predetected anomaly clusters in integral images is one specific example.

In times of machine learning, our model-based approach to mimicking natural swarm behavior (particle swarm optimization) might be deemed old-fashioned and outdated. However, deep learning-based solutions, such as reinforcement learning, have so far failed on our problem. We conjecture that the key reason for this might be the same as for object classification: the unique randomness of occlusions that is specific to a particular operational environment cannot be generalized well by a neural network. The random exploration of PSO, together with a biased exploitation, however, seems to be a simple yet efficient solution. Although the hyperparameters required are well understood[27], we also investigated adaptive PSO variants[54] for automatic parameter estimation. It turned out that the minor additional improvement achieved was not worth the higher processing cost.

Compared to the legacy method, our approach yields a consistent precision gain of about 2–3%, though further trials are needed to confirm

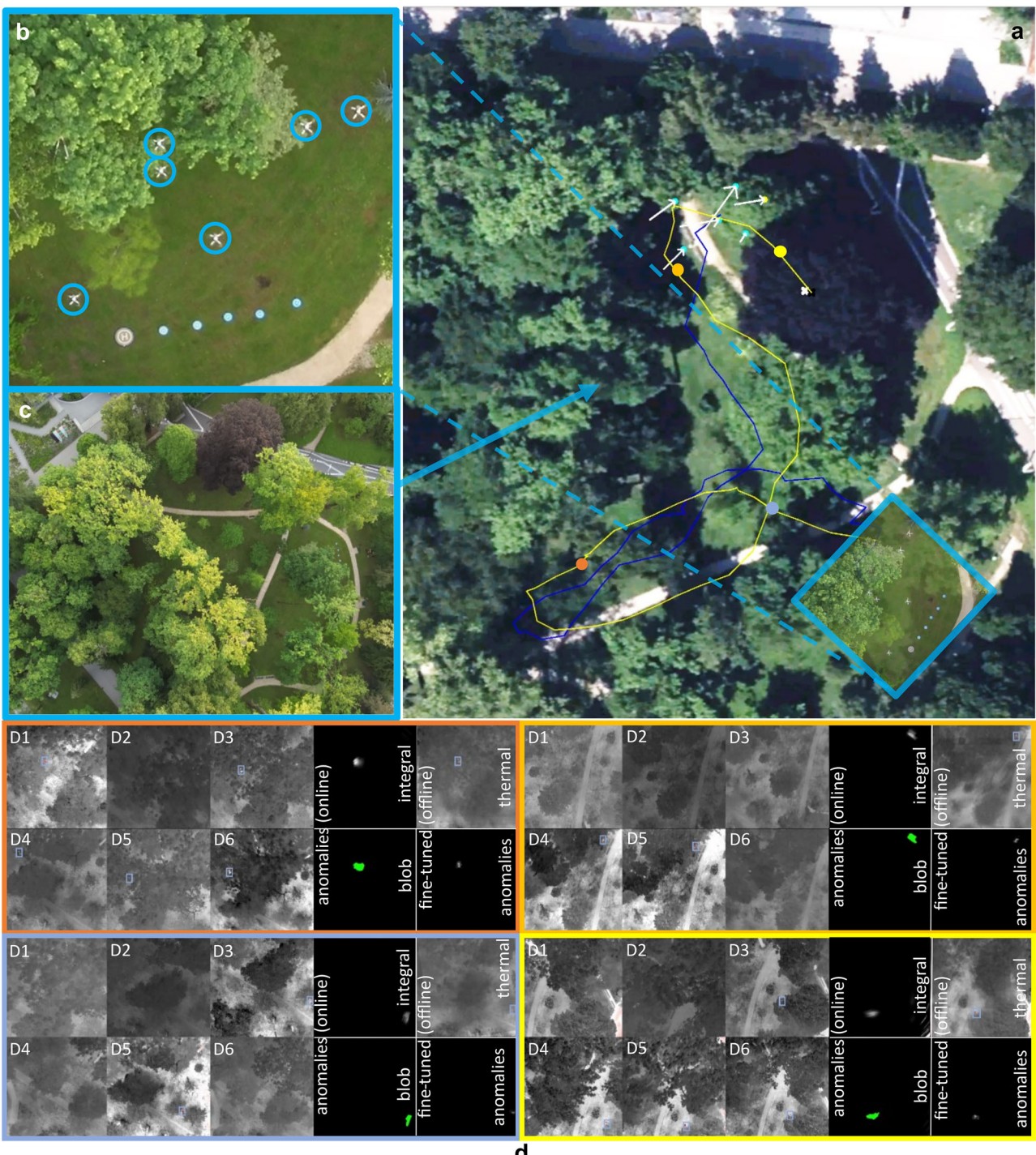

**Fig. 6 | Detection and Tracking of Moving Targets in Dense Forest.** Under difficult conditions (heavy occlusion) the swarm detects and tracks the target that appears most abnormal (in this example, two persons walking side by side). **a** A satellite image with the ground-truth path of the persons (yellow) and the path of the tracking swarm (swarm's center of gravity, blue line), individual drone positions of the swarm at time $t$ (small light blue circles), best sampling position at time $t$ (small yellow circle), and drone movement between time steps $t − 1$ and $t$ (white arrows). **b**, **c** Close-ups (drones encircled). **d** Visual results of single drone images (D1…D6, target positions indicated by boxes), thermal and anomaly integrals, and blobs detected at various waypoints (indicated by circles in **a** and by matching frame colors in **d**). Note that thermal images are color-coded (hot = red, cool = gray). This experiment was recorded in Supplementary Movie 4 *(Experiment III)*.

statistical significance. Importantly, the advantages of our approach extend beyond this small gain. Unlike the legacy method, which requires cluster size–dependent absolute threshold fine-tuning, our method applies a relative, untuned confidence threshold (e.g., $T = 2$), meaning that for a cluster to be deemed the most abnormal, it must appear at least twice as abnormal as the next-most abnormal cluster. This strategy generalizes across different experimental conditions. By also incorporating sensor noise directly into the

synthetic aperture imaging process, it avoids the computational overhead of high-dimensional optimization.

Finally, implementation of sensing drone swarms that go beyond simulation requires a sufficiently fast communication and control hard- and software infrastructure, as well as (especially in the case of SA sensing) extremely high sensor accuracy and precision. Since the accuracy required for image integration through multiple drones cannot be achieved with

today's compass modules, IMUs, pressure sensors, or GNSS, we suggest including all potentially remaining sensor errors in the SA integration process and considering the highest integration correlation to be the correct solution. Only this real-time approach to handling sensor noise enabled us to implement our swarm in practice. This computational solution comes at the cost of larger anomaly clusters compared to those found in perfectly registered integral images and consequently, slightly wrong position estimates. Once again, an additional zero-shot classification of anomaly clusters could improve precision. Note also that while our approach is extendable to arbitrary sensor noise levels, considering larger ranges (e.g., exceeding ±5° heading drift as in our experiments) requires processing additional images over larger sensor ranges and smaller increments. This increases the computational demand and overall processing time.

Although our field experiments included only six drones, we believe that ongoing and rapid technological development will make much larger and faster drone swarms feasible, affordable, and effective in the near future – not only for military but also for numerous civil applications, such as search and rescue, wildfire monitoring, wildlife observation, and forest ecology. We have already shown in simulation that larger swarms are more efficient than smaller ones[27]. With their wider coverage, much faster targets can be tracked, and larger terrain can be searched. See Supplementary Movie 5 *(Conclusion)*. For centralized approaches like ours, large drone swarms demand substantial network bandwidth to stream high-resolution video and telemetry data. While current 5 G networks offer speeds up to 10 Gbit/s, upcoming 6 G technology will support tenfold increases, reaching 100+ Gbit/s. Meanwhile, drone platforms continue to miniaturize—microdrones such as Prox Dynamics' Black Hornet (weighing under 100 g) exemplify ultra-portable designs. Furthermore, flight endurance will be extended through enhancements in hydrogen fuel cells and solar skin technologies. Our processing and imaging speeds are not the bottleneck, as current flight speeds constrain swarm agility. Higher drone velocities will thereby enable tracking of faster-moving targets. However, certain processing tasks—such as per-image anomaly detection—could be decentralized and executed onboard the drones themselves. Currently, our approach detects and tracks only one distinct target deemed most abnormal according to our confidence metric (i.e., at least twice as abnormal compared to the second most abnormal target). If all targets are equally (or very similarly) abnormal, then none of them is detected and tracked. Future work will focus on extending the approach to handle multiple targets simultaneously. With larger swarms, new swarm strategies can be explored that enable the detection and tracking of multiple targets. This necessitates decisions to split the swarm into sub-swarms when targets diverge, and remerge them upon complete target loss (e.g., via model-based optimization like the Hungarian algorithm or learning-based solutions using large reasoning models). With a wider coverage of a larger swarm, much faster-moving targets can also be tracked. Prior simulation[27,55] and statistical analysis[28] demonstrate that occlusion removal improves with higher swarm density through multi-image integration. Specifically, increased sampling enhances dense occlusion removal, though this improvement follows a logarithmic trend rather than scaling linearly. Performance asymptotes to an upper bound determined by occlusion density[28]. This previous research also shows that foliage densities of 50% are most efficient for occlusion removal. The lower the density, the less occlusion removal is necessary. The higher the densities, the more occlusion removal becomes infeasible. Future work will assess the maximum foliage density limits using real LiDAR scans rather than relying solely on simulations or statistical models.

## Materials and Methods

In order to carry out the experiments, we had to implement specific custom hardware and software frameworks that support low-latency transmission and processing of extensive (~70–120 Mbit/s) video and telemetry data (cf. Fig. 7). Since the integration of occlusion and sampling parameters requires individual drone recordings to be registered and averaged to one integral image, a centralized architecture was chosen. We used 6 DJI Mavic 3 T drones and a single DJI Matrice 30 T as platforms for our swarm. These were

chosen because they are equipped with professional thermal and RGB cameras and provide high-quality image stabilization through fast and precise gimbals. A custom application running on DJI Pro / Plus Remote Controllers (implemented with DJI SDK v5.3) has a communication interface for data transmission (telemetry, image/video data, and waypoint information) over a network to a custom Windows server application. The remote controllers and the server PC are interconnected over a switch, thus exchanging data over fast ethernet link. We use the Real-Time Streaming Protocol (RTSP) for streaming the encoded YUV420SP (NV12) video data at 30 Hz. Note that decoding the DJI SDK video stream to extract, re-encode, and transmit only waypoint-relevant frames is significantly slower than transmitting the encoded stream and extracting relevant frames post-transmission and decoding. The telemetry data was embedded in RTSP's AAC audio packet. Furthermore, we use the Message Queuing Telemetry Transport protocol (MQTT) to relay waypoint data. The RCs and the drone communicate via WiFi link, using DJI's proprietary OcuSync protocol. A 24GB Nvidia Geforce RTX 4090 OC GPU on the server was used to decode the video data on its video processing unit (VPU). On average, we achieved a round-trip time (including uploading waypoint data, downloading video and telemetry data, and video decoding) of approx. 80 ms per drone (for up to 10 simultaneously operated drones in the swarm). Two clients were used to connect to the server via a Python wrapper: specifically, 1) a web-based map visualization client (implemented in JavaScript and HTML) that displays each drone's parameters (position, heading, full telemetry, and live video data) on a satellite map, and 2) our swarm control client (implemented in Python 3.7.9), which implements Reed–Xiaoli (RX) anomaly detection[42,43] (see Supplementary Note 2 for background information), image integration (see Supplementary Note 1 for implementation details), blob detection (performed using an OpenCV function[47]), particle swarm optimization, and our objective functions. For our field experiments, we used an autarkic and mobile ground station that can support 10 drone platforms in real time. In addition to a high-end server PC (5.8 GHz Intel i9-13900KF processor (24 cores), 24 GB Nvidia Geforce RTX 4090 OC GPU, 64 GB RAM), a 16x Gigabit switch for fast internal data transmission between remote controllers, PC, and an external 5 G link for networked RTK data transmission was used. A Bosch Power 1500 Prof battery unit provided power in the field for approximately 10 hours. A custom-built handheld networked RTK model (using an Arduino simpleRTK2B v1, a lightweight helical antenna for multiband GNSS, an XBEE bluetooth module, an Android phone running NTRIP Client and Geo Tracker, all integrated into a 3D-printed frame) was employed for ground-truth target tracking. For RTK, the Austrian and German APOS services were used for experiments in Austria and Germany, respectively. More details on our soft- and hardware architecture are provided in Supplementary Note 4 and Supplementary Movie 6 *(Methods)*.

For the swarm's take-off and landing, the following procedure was designed: The drones are arbitrarily positioned on the ground and manually lifted to a holding altitude of approximately. 1 m above ground level. For safety reasons, we neither fly the first nor the last meter automatically or autonomously. The drones subsequently rise automatically to the same (minimum) altitude along the defined linear formation pattern, with the swarm's initial scaling direction (SD) taken from the heading measurement of the center drone (should there be an even number of drones, one of the two middle drones takes on this role). From this holding position, final preflight checks are performed and the initial parameters (e.g., initial SA plane parameters) are set. The drones then rise to their individual relative altitude differences ($\Delta h$) for collision avoidance with other drones in the swarm (as illustrated in Fig. 3C) and start autonomous flight in accordance with the PSO algorithm, as explained above. Landing is achieved by reversing this procedure when the PSO is stopped: From the last PSO formation, the drones return to their take-off positions by maintaining their individual relative altitude differences ($\Delta h$) and descend automatically to the last meter. From here, they are landed manually for safety reasons. Our take-off and landing procedures, as well as the down-wash tests for determining minimum altitude differences versus flight speeds (see

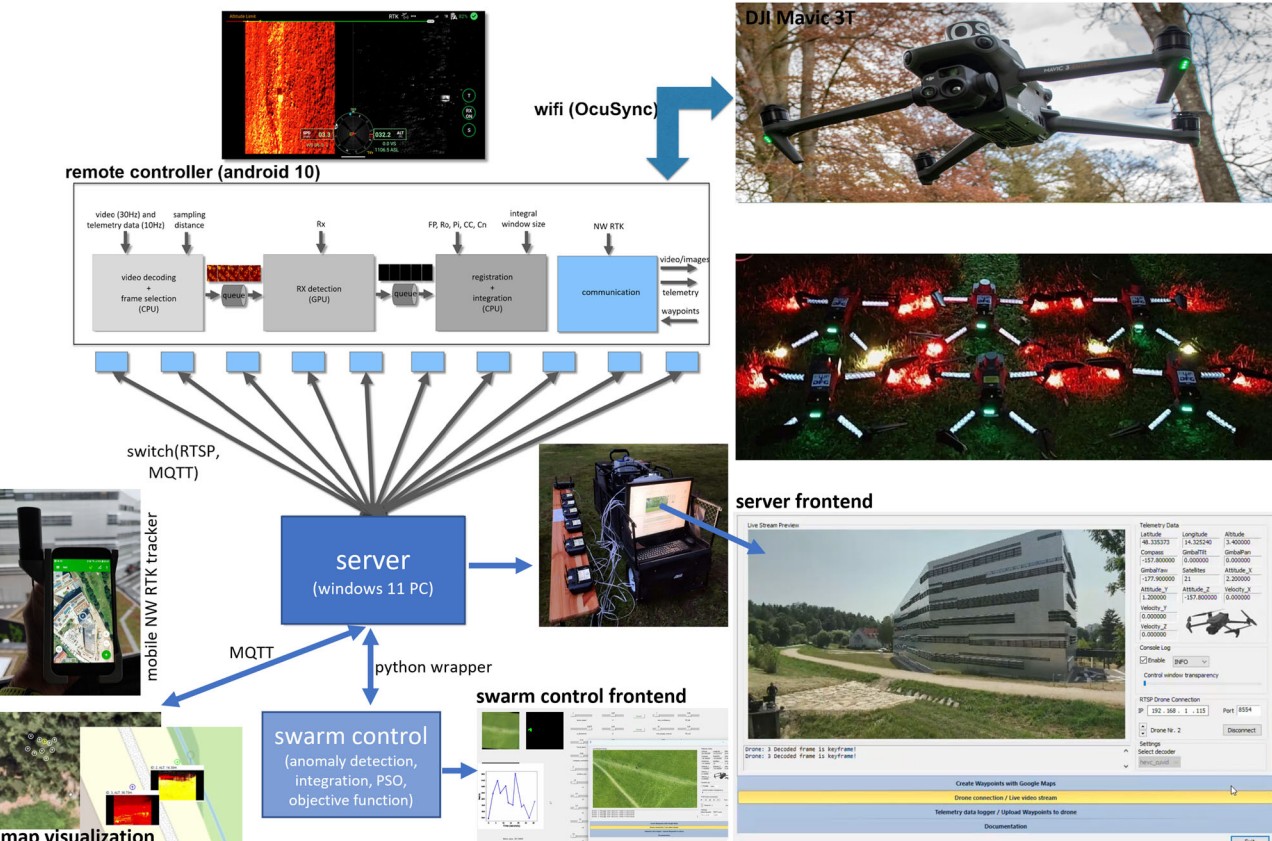

**Fig. 7 | Soft- and Hardware Architecture.** Drones in the swarm are directly controlled by a custom application running on the remote controllers. From the remote controllers, video and telemetry data is down-streamed, and waypoint data is up-streamed to/from a custom server running on a central PC. The data on the server can be downloaded/uploaded by custom clients, such as the swarm control client, which processes the data and controls the swarm (as explained in the main paper), and a map visualization client that visualizes the status of the swarm on a map. See Supplementary Movie 6 (Methods).

Supplementary Note 5 for details), are recorded in Supplementary Movie 6 (Methods).

Our choice of target parameters was based on environmental constraints, sensor parameters, and expected target properties. The maximum tree height, for instance, constrains the minimum flight altitude (approx. 7 m above the canopy in our experiments). The minimum horizontal drone distance $c_4$ depends on the drones' minimum altitude difference $\Delta h$ and the field of view of their cameras. We always chose $\Delta h = 2$ m to ensure a sufficient safety margin in relation to GPS errors. This, together with the selected camera's field of view, fixed $c_4$ to either 4.2 m ($FOV = 43°$ for the RGB camera) or 5.15 m ($FOV = 35°$ for the thermal camera) in our experiments. Since, as explained earlier, $c_1 + c_2 \leq c_4$ and $c_1 \leq c_2$, we chose a 1:2 splitting of $c_4$ into exploration $c_1$ and exploitation $c_2$. Our initial choice for $c_3$ was the expected speed of the target to be tracked. However, as soon as the target is detected, $c_3$ is updated according to its measured speed. The drones' flight speed was chosen such that it was approximately twice the expected maximum target speed (i.e., 3 m/s ≈ 11 km/h for the vehicle in Experiment I, and 2 m/s ≈ 7 km/h for the persons in Experiment III. The speed coefficient $c_5$ for divergence when the target is lost was found experimentally to be $c_5 = 0.3$. The number of drones in the swarm, together with $c_4$ and $\Delta h$, defines the SA's horizontal ($a$) and vertical ($h$) dimensions (cf. Fig. 3C). In our experiments with 6 drones, $a$ was therefore 12.2 m or 15.45 m, and $h$ was 10 m.

The SA focal plane distance $\delta$ is estimated from the altitude above ground level of the drone at $h/2$ (measured using sensor-fused GPS and air pressure), and its orientation ($\theta,\Phi$) is initially set manually by assuming a flat terrain. Note that all three parameters do not have to be estimated exactly, as they are varied during sampling parameter integration. Consequently, height and slope differences in the terrain and sensor errors are taken into account. However, the local terrain visible in one integral image is still considered to be planar. For higher accuracy, a digital elevation model can be applied[21]. To remove outliers, we apply minimum/maximum size and shape constraints by pre-filtering anomaly blobs that (based on our target expectations) are unrealistically small or large, or have inadequate principal components (i.e., shapes). Our swarm control frontend, where these parameters are defined, is shown in Supplementary Movie 6 (Methods).

All materials, including the software of our system and all data of our field experiments, are freely available (see *Data and Materials Availability*).

## Data availability

The data collected for Experiments I, II, and III can be downloaded from Zenodo at: https://doi.org/10.5281/zenodo.13234552. It includes raw single images, raw integral images, anomaly integral images, blob detection results, objective function plots, and classification results. All other data needed to evaluate the conclusions of this paper can be found in the paper or the Supplementary Materials.

## Code availability

Our software system which was used to compute all results presented in this article is available at https://github.com/JKU-ICG/AOS.

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

## Acknowledgements

We thank Patrick Sack at JKU for implementing the map visualization module, Nurullah Özkan and Mohamed Youssef at JKU for helping with the experiments, and JKU science editor Ingrid Abfalter for proofreading the manuscript. This research was funded by the Austrian Science Fund (FWF) and the German Research Foundation (DFG) under grant numbers P32185-NBL and I 6046-N, as well as by the State of Upper Austria and the Austrian Federal Ministry of Education, Science and Research via the LIT-Linz Institute of Technology under grant number LIT2019-8-SEE114.

## Author contributions

O.B. developed the concept, conceived and designed the algorithm and experiments. R.J.A.A.N., S.S., D.M., D.S., and O.B. implemented the algorithm and performed the experiments. R.J.A.A.N., D.M., D.S., and O.B. analyzed the data, contributed materials, and wrote the paper.

## Competing interests

The authors declare no competing interests.
