## [Transparent Peer Review file · Communications Engineering]

An Autonomous Drone Swarm for Detecting and Tracking Anomalies among Dense Vegetation

Corresponding Author: Professor Oliver Bimber

Version 0:

Reviewer comments:

Reviewer #1

(Remarks to the Author)
see attached PDF

Reviewer #2

(Remarks to the Author)
I co-reviewed this manuscript with one of the reviewers who provided the listed reports. This is part of the Communications Engineering initiative to facilitate training in peer review and to provide appropriate recognition for Early Career Researchers who co-review manuscripts.

Reviewer #3

(Remarks to the Author)
This paper presents an autonomous swarm system that detects and tracks anomalies within dense vegetation. The method relies on a parameter-integration approach that folds sensor-noise uncertainty into SA fusion, a PSO-based next-best-view planner, and a custom ground-station network connecting platforms. Experimental results include several scenarios and conditions.

The paper is well-written and organized, and it fits into the scope of the journal. However, this reviewer has some major concerns regarding the paper's contribution and its experimental validation:

- A further discussion and/or evaluation on the scalability of the method in terms of performance and requirements for larger swarms would report significant insight to the manuscript.
- Some parts of the experimental validation should clearly describe the autonomous/semi-autonomous configuration of the evaluation (e.g., Experiment II).
- Real-time / perception rate limits are not analyzed. Particularly, PSO may add >0.5 s. Can this processing rate struggle with fast targets?

Minor changes:

- Typo in Figure 6: "ground-ruth".
- Typo in Page 8: "guidings drone".

Overall, the paper is methodologically correct and reports novel ideas. I do recommend this paper for publication.

Version 1:

Reviewer comments:

Reviewer #1

(Remarks to the Author)

See attached pdf

Reviewer #2

(Remarks to the Author)

I co-reviewed this manuscript with one of the reviewers who provided the listed reports. This is part of the Communications Engineering initiative to facilitate training in peer review and to provide appropriate recognition for Early Career Researchers who co-review manuscripts.

Reviewer #3

(Remarks to the Author)

All my comments have been properly addressed. I do recommend this paper for publication.

Version 2:

Reviewer comments:

Reviewer #1

(Remarks to the Author)

The authors have sufficiently addressed my concerns. I think that the manuscript can be accepted. I recommend an attentive proofreading before publication.

Reviewer #2

(Remarks to the Author)

I co-reviewed this manuscript with one of the reviewers who provided the listed reports. This is part of the Communications Engineering initiative to facilitate training in peer review and to provide appropriate recognition for Early Career Researchers who co-review manuscripts.

An Autonomous Drone Swarm for Detecting and Tracking Anomalies among Dense Vegetation

REVISION

We thank the editor and all reviewers for the valuable comments, which we have addressed as follows:

Reviewers #1 and #2

The two methodological advancements (synthetic aperture fusion and new confidence metric) are neither clearly emphasized nor explicitly presented as novel contributions relative to [27]. The lack of comparative analysis or ablation study leaves the methodological progression underexplored and the practical gains unquantified. This weakens the reader's ability to appreciate the full scope of the improvements achieved in the current work.

While sensor noise integration and the confidence metric are now highlighted in the abstract and introduction as contributions, this work also makes significant additional advances: the hardware implementation of our autonomous swarm strategy (requiring new hardware/software infrastructure) and its evaluation in field experiments. We hope reviewers recognize the importance of these contributions, as many robotics studies present simulation-only results, thereby avoiding the challenges inherent in real-world deployment.

Please note that [27] presented pure simulations without sensor noise. We have already demonstrated—both in simulation and using real-world sensor data—the detrimental effect of sensor noise (see Fig. 2 and discussion). Without noise integration, sensor noise causes misregistered images and prevents reliable anomaly detection.

The legacy confidence metric in [27] identified the most abnormal target based on the maximal contour size of connected pixel clusters in the integrated anomaly masks. This approach equated the largest spatial anomaly in the integral image with the highest abnormality—a fundamental limitation, as contour size reflects only spatial extent, not true anomaly severity. Our proposed metric addresses this by integrating the anomaly values over the target area, thereby jointly considering spatial scale and anomaly intensity. Consequently, smaller targets exhibiting stronger anomalies can now be prioritized over larger ones. We explain this now in the section Swarm Formation and Sampling.

We also re-evaluated the recordings from all three experiments using our legacy method [27] (ignoring sensor noise and using the old confidence metric). To compare the approaches, we optimally selected the contour size threshold for the old metric to achieve 100% recall. Precision dropped from 93.9% to 91.8% for Experiment I and from 92.3% to 89.3% for Experiment III. Note that the precision loss depends on the amount of sensor noise, particularly compass drift over time. Thus, precision would be further degraded by sensor noise exceeding the levels observed during our experiments. For Experiment II, neither the lying nor standing person was detected

with the legacy method (only false positives). We now present these numbers in the corresponding experiment sections.

In this study, the approach seems to be genuinely centralised. This should be clearly stated from the beginning.

We now clearly state (additionally in the abstract as well as in the introduction) that our approach (i.e., the collection and integration of the wide SA optical signal) requires a centralized architecture.

The abstract and introduction assume expert-level familiarity with core concepts such as “synthetic aperture” and “particle swarm optimization,” without first introducing or contextualizing them.

We would like to leave the abstract on this level, as synthetic aperture sensing/imaging and particle swarm optimization are classical topics in signal processing, optimization and robotics. We have provided appropriate citations to prior work on these topics.

In addition, placing the “Materials and Methods” section before the conclusion breaks conventional article structure and hinders clarity.

Placing the Materials and Methods section after Results (not the conclusions - they are provided in the Discussions section after Materials and Methods) aligns with the standard structural convention of almost all Nature and Science journals. This organization enables readers to focus first on results without initial technical barriers. We therefore would like to maintain this structure, consistent with our prior work [27] where we also applied this order (also published in *Communications Engineering*).

The system uses video input from UAVs and treats each frame independently (as static images), but does not explain why continuous video is necessary over discrete image sampling. If temporal information is exploited, this should be made explicit and evaluated; otherwise, image capture could simplify the pipeline without loss of performance. This needs to be clarified.

We now explain in the Materials and Methods section the reason: The decoding of the DJI SDK video stream to extract, re-encode, and transmit only waypoint-relevant frames is significantly slower than transmitting the encoded stream and extracting relevant frames post transmission and decoding. So we lose performance if we transmit only the images due to decoding and encoding. In fact, we started with this approach, but it proved to be far too slow.

The limitations of the method under high occlusion or close-range conditions are not quantified. Operational boundaries such as maximum effective foliage density, minimum detection range, minimum distance between foliage and target, or blob confidence thresholds are not discussed, which would be crucial for real-world deployment.

We now explain in the discussion section that prior simulation [27] and statistical analysis [28] demonstrate that occlusion removal improves with higher swarm density through multi-image integration. Specifically, increased sampling enhances dense occlusion removal, though this improvement follows a logarithmic trend rather than scaling linearly. Performance asymptotes to an upper bound determined by occlusion density [28]. The choice of drone distances and altitudes was discussed in [27] and was summarized in the section Swarm Formation and Sampling. We are not sure what you mean by minimum distance between foliage and target. In case you enquire about the minimum distance between foliage and drones, then, as discussed in the paper, this should be as low as possible. For the choice on the confidence threshold we now explain in the Swarm Formation and Sampling section that we determined empirically $T=2$ (i.e., a cluster to be considered the most abnormal, it must appear at least twice as abnormal as the second-most abnormal cluster) to be suitable for all experiments.

The role of YOLO is ambiguous in Experiment I: its inclusion, class selection, and lack of integration into the main method are insufficiently justified.

Note that object classification (here implemented with YOLO) is not part of our framework. We apply it solely as a benchmark to demonstrate its limited effectiveness in complex forested terrain with partial occlusion – consistent with prior findings [18]. For this evaluation, we restricted classes to vehicles and persons, reflecting our target application scenario. This restriction prevents false detections from irrelevant classes and represents the optimal configuration for this task. Nevertheless, performance remains poor (Table S1). We explain this better in the explanation of Experiment 1 now. Note also, that our approach does not perform classification. Anomaly detection is therefore independent of detected classes. Hence, if the most abnormal target is a person, it would be detected.

The guided search is presented without timing data, confidence progression, or comparison to unguided baselines, making it more a proof of concept than a fully developed experiment.

To address potential misunderstandings regarding the term “guided search”, we have reorganized the Experiment II section. For context: Experiment I focuses on detecting and tracking moving targets in sparse forest, Experiment II addresses localized target detection in a dense forest, and Experiment III involves detecting/tracking of moving targets in dense forest. Guidance was solely employed for initial swarm deployment within the region of interest and does not impact our sampling methodology. Supplementary tables and plots S1 and S3 demonstrate moving target

tracking over time for Experiments I and III. As Experiment II deals with static targets where tracking over time is inapplicable, we instead provide distance precision relative to ground truth positions for both standing and lying person detections. Further clarification on the “guided search” terminology and our revisions follows below.

For failure cases in Experiment III, no analysis is given for when or why detections failed.

We now provide an explanation in Experiment III: While false negative detections result from temporary invisibility (e.g., due to locally dense vegetation), false positive detections occur when extraneous features appear more anomalous than the actual target, such as larger hotspots on the ground.

Figure S1: we suggest enhancing this with an explicit schematic or pseudocode block that walks through the SA integration logic

We have added the pseudocode that explains Figure S1, and how integral images are computed.

YOLO benchmarking: The inclusion of only 'vehicle' and 'person' classes is not well justified, particularly since no humans appear in this scenario. Several rows in Table S1 are marked as correct detections despite the absence of YOLO scores.

As noted above, this benchmark demonstrates a hypothetical use case for detecting vehicles and persons in forested terrain. However, it does not presuppose human presence. We have clarified this purpose in the revised Experiment I description. Crucially, our anomaly detection method performs no classification; it identifies the most salient anomaly regardless of class (e.g., a person exhibiting abnormal behavior would be detected). In this experiment, the primary anomaly was the vehicle.

Please note that column #6 is the detection of our approach (not YOLO), while column #7 is the YOLO confidence score. A missing confidence score indicates that YOLO did not detect the target at all (not even with a low confidence score). So an entry “correct” in column #6 and no confidence score in column #7 means that we detect the target correctly with our approach, while YOLO does not detect it at all and therefore does not provide any confidence score. However, even low YOLO confidence scores will lead to a detection failure unless the confidence score threshold is chosen very low - which then leads to many false positives. We added this information in the caption of Table S1.

Guided search: The paper lacks detail on how guidance is performed (e.g., what the operator sees, how its termination triggers the autonomous phase). Without such detail, it is unclear how operator actions influence fleet initialization or coverage.

We acknowledge that our original term "guided search" may have caused confusion. This process is solely for the initial deployment of the swarm to a specific region of interest prior to the commencement of autonomous search operations. It does not impact the swarm's autonomous behavior itself. This initial deployment could have been achieved through alternative methods, such as defining the ROI via a ground station map, rather than having the swarm follow a dedicated guidance drone to a hover point. During this phase, the operator views the guidance drone's live feed on the remote controller while monitoring the swarm's processed results on a separate PC display. Crucially, there is no algorithmic connection or direct correlation between these two distinct video streams. To reiterate, the guidance drone serves only to position the swarm efficiently and directly at the designated ROI before autonomous search begins. Without this initial positioning, the swarm's autonomous search strategy might not locate the specific ROI effectively. To address the potential misunderstanding, we have reorganized the Experiment II section (which focuses on detecting localized targets in dense forests, while Experiment I is about detecting and tracking moving targets in sparse forest, and Experiment III is about detecting and tracking targets in dense forest).

Inconsistency in failure cases Table S3 vs Figure S3 (4 vs 8).

There is no inconsistency between the results shown in Figure S3 and Table S3. The 4 locations in the plot of Figure S3 where the confidence drops (below our threshold) are cases where we do not detect the target at all (i.e., false negatives). These are indicated with "no" in column #6 of Table S3 (exactly 4). The cases indicated with "wrong" in Table S3 are false positives (detected by our approach, but wrong). We added this to the captions of Tables S3 and S1.

Discuss system limitations in conclusion (e.g., centralized computation, no onboard intelligence, false positives, scalability concerns) and more concrete suggestions for future work (e.g., multi-target tracking, online failure recovery, domain adaptation for anomaly detectors).

We extended the Discussion section to address technical requirements for scaling to larger swarms—noting that while simulation studies in our preceding work [27] demonstrated advantages of larger swarms (wider coverage, faster target tracking, and expanded search area), real-world deployment necessitates advances in network bandwidth, platform portability, and flight endurance.

Furthermore, we expanded the discussion on false positive limitations and anomaly detection research by exploring environmentally adaptive deep-learning detectors. For multi-target tracking, we proposed concrete strategies including model-based approaches (e.g., Hungarian algorithm) and learning-based solutions using large reasoning models to dynamically split/merge swarms as targets diverge/converge.

Although our centralized architecture remains essential for integrating all drone-captured images into a unified composite, we note that certain processing tasks (e.g., per-image anomaly detection) could be decentralized to onboard systems. However, computational processing—and

with sufficient bandwidth, data transmission—does not constrain system agility. The primary limitation stems from drone flight speeds, where higher velocities would enable greater swarm responsiveness.

Reviewer #3

A further discussion and/or evaluation on the scalability of the method in terms of performance and requirements for larger swarms would report significant insight to the manuscript.

We have evaluated our approach's performance with increasing swarm sizes in simulation (as detailed in our preceding work [27]), demonstrating advantages of larger swarms—including wider coverage, faster target tracking, and expanded searchable terrain. Therefore, we have extended the scalability discussion in the Discussion section to address critical technological limitations requiring solutions for supporting scalability, such as network bandwidth, platform portability, and flight endurance.

Some parts of the experimental validation should clearly describe the autonomous/semi-autonomous configuration of the evaluation (e.g., Experiment II).

This point was also brought up by reviewers #1 and #2. We have addressed it as follows: We acknowledge that our original term "guided search" may have caused confusion. This process is solely for the initial deployment of the swarm to a specific region of interest prior to the commencement of autonomous search operations. It does not impact the swarm's autonomous behavior itself. This initial deployment could have been achieved through alternative methods, such as defining the ROI via a ground station map, rather than having the swarm follow a dedicated guidance drone to a hover point. During this phase, the operator views the guidance drone's live feed on the remote controller while monitoring the swarm's processed results on a separate PC display. Crucially, there is no algorithmic connection or direct correlation between these two distinct video streams. To reiterate, the guidance drone serves only to position the swarm efficiently and directly at the designated ROI before autonomous search begins. Without this initial positioning, the swarm's autonomous search strategy might not locate the specific ROI effectively. To address the potential misunderstanding, we have reorganized the Experiment II section (which focuses on detecting localized targets in dense forests, while Experiment I is about detecting and tracking moving targets in sparse forest, and Experiment III is about detecting and tracking targets in dense forest).

Real-time / perception rate limits are not analyzed. Particularly, PSO may add >0.5 s. Can this processing rate struggle with fast targets?

Processing and imaging speeds (hardware-dependent) are not the limiting factor for swarm agility, as current flight velocities primarily constrain reconfiguration times. While imaging and processing occur on millisecond timescales, flight maneuvers require seconds. Consequently, higher drone velocities would enable tracking of faster-moving targets. We have incorporated this discussion in the Discussion section. This extends our preceding work [27], where we simulated performance with increasing target speeds.

Minor changes:

- ***Typo in Figure 6: "ground-ruth".***
- ***Typo in Page 8: "guidings drone".***

Fixed.

Overall, the paper is methodologically correct and reports novel ideas. I do recommend this paper for publication.

An Autonomous Drone Swarm for Detecting and Tracking Anomalies among Dense Vegetation

REVISION

We thank the editor and all reviewers for the valuable comments, which we have addressed as follows:

The Materials and Methods section has been moved to the end of the manuscript to align with the standard Communications Engineering structure.

Reviewers #1 and #2

Experiment I

1). The benchmarking against YOLO should be better described. In particular, having access to images from different drones, it should be specified which image was used for object detection: only one, all of them or something in between? It is not specified what is the confidence measure used for YOLO. Also, there is confusion between YOLO confidence and the confidence used for the proposed approach, which are clearly two different things, and should be named in a different way.

In Experiment 1, object detection with YOLO-World v2 was performed on the image captured from the drone perspective corresponding to the best sampling position (P_{best}^t). This position is defined as the viewpoint for which the computed integral I_{best}^t , representing our confidence measure c , is maximal. This perspective not only provides the best viewpoint but also yields the highest YOLO confidence score among all images captured by the swarm at time t . Therefore, we report the YOLO confidence score obtained from this best drone perspective.

Please note that “confidence score” is a standard term in machine learning classification, but its exact definition can vary between models. The confidence score used for YOLO-World v2 corresponds to the model output as described in [52], whereas our confidence score is defined by the objective function presented in this paper. To avoid ambiguity in naming between the two, we now explicitly refer to them as “YOLO confidence score” and “our confidence score”.

We now add the following to the Experiment I section:

“Object recognition was carried out on the image captured from the drone perspective corresponding to the best sampling position P_{best}^t , i.e., the viewpoint for which the computed integral I_{best}^t , representing our confidence measure c , is maximal. This viewpoint also yielded the highest YOLO confidence score among all images captured by the swarm at time t .“

2). Benchmarking against the legacy method is important to understand the actual contribution, but it would be useful to specify that this is performed in an open loop (much as with YOLO), just using the images taken by the drones during their flight. Results should be added to table S1 as well.

The benchmarking against the legacy method is indeed performed in an open-loop manner, using only the images captured by the drones during their flights, as done with YOLO for automatic classification.

We now add the following to the Experiment I section:

“Re-evaluating against the legacy method ^[27] *in an open-loop manner, using only the images captured by the drones as done with YOLO for automatic classification* (ignoring sensor noise and using its original confidence metric) shows a precision drop from 93.9% to 91.8%.”

We have made the following change in the supplementary material:

Table S1 has been updated to facilitate a comprehensive comparison; we have added a detection column for the legacy method in Table S1, including both precision and recall, alongside the existing detection results from our approach.

3). It is not clear if the reduction in precision with the legacy method can be considered statistically significant or not.

The observed difference represents a small improvement of approximately 2 to 3% in precision with our approach compared to the legacy method. We now acknowledge this in the Discussion section, noting that while the improvement is consistent across our experiments, additional trials will be required in the future to conclusively establish statistical significance.

We would also like to emphasize that the advantages of our approach extend beyond this 2–3% precision gain. In the legacy method, the absolute threshold must be fine-tuned to achieve the best possible results (e.g., an optimal threshold configuration that yields 100% recall), which is not practical in real-world scenarios. By contrast, our approach employs a relative and flexible (i.e., untuned) confidence threshold T . For instance, $T=2$ (empirically determined to work well across all experiments) requires that, for a cluster to be considered the most abnormal, it must appear at least twice as abnormal as the second-most abnormal cluster. In the legacy method, however, the absolute threshold must be adjusted according to the target cluster size; smaller clusters require a lower absolute threshold, while larger clusters require a higher one. Therefore, our approach is more general and can be applied consistently across different cluster sizes and experimental conditions. Furthermore, unlike the legacy method, our approach incorporates sensor noise into the synthetic aperture imaging process, eliminating the need for computationally expensive optimization over high-dimensional parameter spaces.

We now add the following to the Discussion section:

“Compared to the legacy method, our approach yields a consistent precision gain of about 2–3%, though further trials are needed to confirm statistical significance. Importantly, the advantages of our approach extend beyond this small gain. Unlike the legacy method, which requires cluster size–dependent absolute threshold fine-tuning, our method applies a relative untuned confidence threshold (e.g., $T=2$), meaning that for a cluster to be deemed the most abnormal, it must appear at least twice as abnormal as the next-most abnormal cluster. This strategy generalizes across different experimental conditions. By also incorporating sensor noise directly into the synthetic aperture imaging process, it avoids the computational overhead of high-dimensional optimization.”

Experiment II

4). The accompanying video indicates that the two thermal targets (one standing, one lying) are located in separate forest regions, and the 7th “deployment” drone flies to each location sequentially to initialize swarm exploration. This spatial separation is not clearly stated in the manuscript, yet it significantly affects how the swarm detects and tracks anomalies. It appears that at each deployment location, only one target is within the swarm’s sensing footprint, but no details are provided on whether the second target was outside the field of view or ignored by the system. Please clarify.

We clarify that the two thermal targets (i.e., one standing and one lying person) are indeed located in separate forest regions, corresponding to distinct deployment locations. At each deployment location, only one target falls within the swarm’s sensing footprint, while the second target is outside the sensing range and is therefore not detected at that time.

We now add the following to the Experiment II section:

“The targets were located in distinct deployment regions, such that at each deployment location only one target fell within the swarm’s sensing footprint, while the other target remained outside the sensing range.”

5.) It is not clear why the automatic classification is unrealistic here, and why manual confirmation is necessary. In the end, if the threshold $T=2$ is overcome, then a blob gets detected also in this experiment. Why not use this method?

We believe there may be a misunderstanding regarding the distinction between automatic detection and automatic classification. In this experiment, the confidence threshold $T=2$ is used for automatic detection of anomaly blobs, which reliably identifies regions of interest (i.e., potential targets). However, automatic classification, for example labeling detected blobs with a model such as YOLO, requires sufficiently clear image features to accurately recognize and categorize the target class, such as identifying a blob as a person. In scenarios with extreme occlusion, these features are often insufficient or obscured, making automatic classification unreliable. A similar limitation was observed even under sparse occlusion in Experiment 1. Therefore, manual confirmation, that is classification by a human operator, is necessary to validate detections based on thermal and anomaly integrals in real-world experiments where no ground truth information is available.

We have now provided a clearer explanation of this in the Experiment II section:

“Note that, after successful detection, a reliable automatic classification (e.g. using a classifier such as YOLO) under such extreme occlusion conditions is not realistic (see Experiment I), manual classification by a human operator by visually observing the thermal and anomaly integrals computed might be more efficient (cf. Figure 5C, F).”

6). Why in table S2 are you providing results for a single detection and not a sequence? The swarm was anyway exploring after the deployment drone stopped, hence there should be a sequence of anomaly integrals much as in Experiment I.

Unlike Experiments 1 and 3, the objective of this Experiment is to detect stationary targets whose locations do not change over time. Since the task here is primarily detection rather than continuous tracking, a single successful detection is deemed sufficient to confirm the presence of the anomalous target. Any additional successful detections would be redundant and would not provide further insights. Therefore, Table S2 focuses primarily on the results of that single detection.

We now add the following to the supplementary material:

We have now added a table (Table S3), showing the sequence of detection results for Experiment II using our approach and of the legacy method to facilitate a comprehensive comparison.

We have also added the following to Experiment II section:

“In contrast to tracking moving targets (see Experiment I and III) detecting stationary targets requires only one correct detection after deployment. This is the case for our new approach in this experiment while the legacy method fails.”

7). How is it possible that the legacy method did not provide any blob for detection? Given that blobs exist in the anomaly integral, also the legacy method should detect them, although maybe giving more importance to non-target objects. Please provide full data for both the proposed method and the legacy approach.

It is important to note that in this experiment a single correct detection is sufficient to confirm the presence of an anomalous target as the objective here is primarily detection rather than continuous tracking. Please refer to comment 6.

The legacy method relies on an absolute threshold that must be fine-tuned to achieve the best possible results (e.g., an optimal threshold configuration that yields 100% recall). This fine-tuning is impractical in real-world scenarios, as the absolute threshold must vary depending on the size of the target cluster: smaller clusters require lower absolute thresholds, while larger clusters require higher ones. Although blobs are present in the anomaly integral, the legacy method did not produce any true positive detections for any chosen absolute threshold. In other words, neither the lying person nor the standing person was successfully

detected. Depending on the selected absolute threshold, the legacy method may register false positives, but it consistently failed to detect the actual targets and therefore did not meet the experiment objective.

In contrast, the proposed approach which is more general, employs a relative and flexible (i.e., untuned) confidence threshold T . For instance, $T=2$ (empirically determined to work well across all experiments) requires that, for a cluster to be considered the most abnormal, it must appear at least twice as abnormal as the second-most abnormal cluster. This strategy enables successful detection of heavily occluded targets, including both the lying and standing person, and demonstrates robustness across different cluster sizes and experimental conditions. Therefore, providing detailed detection data for the legacy method may not yield additional meaningful insight. The proposed method, however, clearly demonstrates its effectiveness in reliably detecting targets under challenging conditions.

As mentioned under comment 6, we now added the following to the supplementary material:

We have now added a table (Table S3), showing the sequence of detection results for Experiment II using our approach and of the legacy method to facilitate a comprehensive comparison.

We have also added the following to Experiment II section:

“In contrast to tracking moving targets (see Experiment I and III) detecting stationary targets requires only one correct detection after deployment. This is the case for our new approach in this experiment while the legacy method fails.”

8). Given the stated manual detection confirmation step, how were the errors computed? It would be valuable to quantify the detection effort, including time to detection, swarm path or motion (e.g., center of mass trajectory), and the stopping criterion used to confirm detection.

We believe this may be a misunderstanding, and it is also related to our response to comment 5. The detection of anomaly blobs, which reliably identifies regions of interest (i.e., potential targets) is fully automatic. The statement regarding the manual classification confirmation step pertains specifically to real-world experiments where ground truth data is unavailable. In such cases, a human operator may provide validation, as automatic classification methods (e.g., YOLO) fail completely under conditions of dense occlusion. For the experiments reported in this work, errors were computed by comparing the automatically detected anomaly blobs against the ground truth target locations.

As mentioned in comment 5, we have now better explained this in the Experiment II section:

“Note that, after successful detection, a reliable automatic classification (e.g. using a classifier such as YOLO) under such extreme occlusion conditions is not realistic (see Experiment I), manual classification by a human operator by visually observing the thermal and anomaly integrals computed might be more efficient (cf. Figure 5C, F).”

Experiment III

9). Why have you chosen two persons walking side by side as a target? Is this because the anomaly of a single person would not be recognisable?

We chose two persons walking side by side as a target to address different use cases. This choice is not due to difficulties in detecting single-person anomalies, which, as shown in Experiment 2, can also be reliably detected.

We now add the following to the Experiment III section:

“We chose two persons walking side by side as target in this experiment to extend our use cases over single person targets (Experiment II) and vehicles (Experiment I).”

10). It is not clear if the reduction in precision with the legacy method can be considered statistically significant or not.

Please refer to comment 3 for a detailed response on this point.

11). How could you obtain 100% recall for the legacy method (no false negatives) if the proposed approach fails 4 times because of “temporary invisibility (e.g., due to locally dense vegetation)”?

As mentioned earlier in comment 3 and comment 7:

In the legacy method, the absolute threshold used was fine-tuned to achieve the best possible results (e.g., an optimal threshold configuration that yields 100% recall). This setting reduces false negatives to zero but with a potential increase in the number of false positives. Note, such fine-tuning is impractical in real-world scenarios, since the required absolute threshold varies with the size of the target cluster: smaller clusters require lower absolute thresholds, while larger clusters require higher ones.

By contrast, our approach employs a relative and flexible (i.e., untuned) confidence threshold T . For instance, $T=2$ (empirically determined to work well across all experiments) requires that, for a cluster to be considered the most abnormal, it must appear at least twice as abnormal as the second-most abnormal cluster. Because this confidence threshold is not overfit to each case, the proposed approach can occasionally miss targets (e.g., temporary invisibility due to locally dense vegetation), which explains the 4 “failures.” Unlike the legacy method, which requires adjusting absolute thresholds according to cluster size, our method generalizes consistently across different cluster sizes and experimental conditions.

We now add the following to the supplementary material:

“False negatives are absent for the legacy method because the absolute threshold was fine-tuned to achieve the best possible results (i.e., an optimal threshold configuration that yields 100% recall). Note, that such fine-tuning is unrealistic under real-world conditions as anomaly cluster are unknown.”

12). Please provide in Table S3 all the results of the legacy method as well.

In the revised version, Table S3 is now labeled as Table S4.

We have made the following change in the supplementary material:

Table S4 is updated to facilitate a comprehensive comparison; we have added a detection column for the legacy method, including both precision and recall, alongside the existing detection results from our approach.

Quantitative Boundaries on Occlusion Limits

13). The following factors remain unquantified:

i). The maximum level of foliage density at which anomaly detection becomes unreliable.

We have examined this question in our previous work using both a statistical model [28] and a simulated forest model [55], as well as in swarm scenarios [27]. Our findings indicate that detection performance is not a binary function of foliage density, but depends on both density and sampling frequency as explained in the Discussion section. Across all three studies [27-28,55], occlusion removal was most effective at around 50% foliage density; at lower densities, it is less necessary, while at higher densities it becomes increasingly infeasible. We would like to emphasize that this is a general AOS topic rather than a specific focus of the current paper. As part of future work, we plan to assess these limits using real LiDAR scans rather than relying solely on simulated forests or statistical models.

The Discussion already stated the following in the previous revision:

Prior simulation [27,55] and statistical analysis [28] demonstrate that occlusion removal improves with higher swarm density through multi-image integration. Specifically, increased sampling enhances dense occlusion removal, though this improvement follows a logarithmic trend rather than scaling linearly. Performance asymptotes to an upper bound determined by occlusion density [28]. This previous research also shows that foliage densities of 50% are most efficient for occlusion removal. The lower the density, the less occlusion removal is necessary. The higher the densities, the more occlusion removal becomes infeasible.

We now add the following to the Discussion section:

“This previous research also shows that foliage densities of 50% are most efficient for occlusion removal. The lower the density, the less occlusion removal is necessary. The higher the densities, the more occlusion removal becomes infeasible. Future work will assess the maximum foliage density limits using real LiDAR scans rather than relying solely on simulations or statistical models.”

ii). The system's sensitivity to sensor orientation errors, particularly heading drift beyond the stated $\pm 5^\circ$ tolerance.

Our approach is extendable to all levels of sensor noise. However, for sensor errors exceeding the specified $\pm 5^\circ$ tolerance (e.g., heading drift), compensation requires processing additional images over larger ranges and smaller increments. This adjustment may increase computational demand and overall processing time.

We now add the following to the Discussion section:

“Note also, that while our approach is extendable to arbitrary sensor noise levels, considering larger ranges (e.g., exceeding $\pm 5^\circ$ heading drift as in our experiments) requires processing additional images over larger sensor ranges and smaller increments. This increases the computational demand and overall processing time.”

14). How does the system behave in presence of multiple, similarly salient targets? Would the algorithm alternate between them, reject both, or lock onto one arbitrarily? A brief discussion acknowledging these limitations, and their implications for multi-target scenarios, would significantly enhance the transparency and interpretability of this experiment.

Our current approach is designed to detect and track only a single target, which is identified as most abnormal by our confidence metric. When multiple targets have similar abnormality scores, the single-target objective fails and none of the targets are detected or tracked.

We now add the following to the Discussion section:

“Currently, our approach detects and tracks only one distinct target deemed most abnormal according to our confidence metric (i.e., at least twice more abnormal compared to the second most abnormal target). If all targets are equally (or very similarly) abnormal then none of them is detected and tracked. Future work will focus on extending the approach to handle multiple targets simultaneously.”

Reviewer #3

All my comments have been properly addressed. I do recommend this paper for publication.

An Autonomous Drone Swarm for Detecting and Tracking Anomalies among Dense Vegetation

Reviewers #1 and #2

1). I recommend an attentive proofreading before publication.

We have carefully proofread the manuscript and corrected any errors.

Review of "An Autonomous Drone Swarm for Detecting and Tracking Anomalies among Dense Vegetation"

Summary Evaluation

The paper presents an autonomous multi-drone architecture using anomaly-based detection and synthetic aperture integration for tracking occluded targets. The method employs a confidence-driven PSO strategy to coordinate drones and fuses multiple anomaly detections across uncertain sampling parameters. Through three progressively challenging real-world experiments, the method shows strong detection accuracy and system robustness. It offers a valuable contribution toward real-world deployment of autonomous aerial sensing through detailed description of the system. However, some evaluation and description aspects could be improved.

Strengths

- Integration of anomaly detection with synthetic aperture imaging is conceptually elegant and practically validated.
 - PSO-based control is well-motivated and adaptation of PSO is well explained and shown to be robust to environmental complexity.
 - Experiments are conducted in real-world conditions with increasing complexity.
 - Evaluation includes spatial accuracy, confidence metrics, and qualitative illustrations.
-

Major Weaknesses

- The paper builds on prior work [27], which introduced the core PSO-based anomaly detection and synthetic aperture fusion framework in simulation. The current work implements the system on real drones, a non-trivial challenge, but does not extend the methods significantly. We recognise that the present study introduces two methodological enhancements that emerged from adapting the system for real-world deployment: (i) synthetic aperture fusion over discretized heading, tilt, and focal plane parameters to handle pose uncertainty, as perfect position and orientation data are no longer available; and (ii) a confidence metric, defined as the ratio between the most and second-most salient anomaly blobs, in contrast to the single-blob objective used previously, used to control transitions between exploration and convergence behaviors. The core of the methodology remains unchanged. Additionally, these two methodological advancements are neither clearly emphasized nor explicitly presented as novel contributions relative to [27]. The lack of comparative analysis or ablation study leaves the methodological progression underexplored and the practical gains unquantified. This weakens the reader's ability to appreciate the full scope of the improvements achieved in the current work.

- The term “swarm” is used extensively, yet the implementation is fully centralized with respect to data fusion and control, as acknowledged in the materials section. This contradicts the notion of a decentralized autonomous swarm described in the introduction. In the swarm robotics domain, centralised approaches are often dismissed, as they contrast the concept of autonomy of the individual units and the self-organisation of the system behaviour. While experiments in swarm robotics are often conducted with some central entity providing key information (e.g., position of neighbours), this is done to compensate for the lack of sensory devices that can provide the required information. In this study, instead, the approach seems to be genuinely centralised. This should be clearly stated from the beginning.
 - The abstract and introduction assume expert-level familiarity with core concepts such as “synthetic aperture” and “particle swarm optimization,” without first introducing or contextualizing them. In addition, placing the “Materials and Methods” section before the conclusion breaks conventional article structure and hinders clarity. The paper structure and clarity should be significantly improved.
 - The system uses video input from UAVs and treats each frame independently (as static images), but does not explain why continuous video is necessary over discrete image sampling. If temporal information is exploited, this should be made explicit and evaluated; otherwise, image capture could simplify the pipeline without loss of performance. This needs to be clarified.
 - The limitations of the method under high occlusion or close-range conditions are not quantified. Operational boundaries such as maximum effective foliage density, minimum detection range, minimum distance between foliage and target, or blob confidence thresholds are not discussed, which would be crucial for real-world deployment.
 - The role of YOLO is ambiguous in Experiment I: its inclusion, class selection, and lack of integration into the main method are insufficiently justified.
 - Experiment II, while conceptually interesting, lacks quantitative support. The guided search is presented without timing data, confidence progression, or comparison to unguided baselines, making it more a proof of concept than a fully developed experiment.
 - For failure cases in Experiment III, no analysis is given for when or why detections failed.
-

Section-by-Section Notes

Synthetic Aperture & PSO (Methods)

A key strength is the synthetic aperture (SA) integration approach, which fuses anomaly masks across a discretized range of heading, tilt, and focal plane parameters. This allows the system to robustly handle pose uncertainty without requiring optimization, as anomaly signals reinforce while misalignments cancel out. Figure S1 illustrates how anomaly masks are fused through Airborne Optical Sectioning (AOS), helping to clarify the effects of different SA parameters. This visualization partially addresses the need for interpretability, though it still requires attentive reading. Given that integrating sensor noise and sampling uncertainty into the synthetic aperture

fusion is central to the method, we suggest enhancing this with an explicit schematic or pseudocode block that walks through the SA integration logic. This would significantly improve clarity for technically engaged readers and support reproducibility.

Complementing this, the multi drone system adapts dynamically using a confidence-driven PSO objective function. The transition between search and convergence modes is well described, and the system geometry is adjusted to reduce occlusion and inter-drone interference. Confidence is defined cleanly as a salience ratio between dominant and subdominant anomaly blobs. PSO logic and the adaptation are clearly explained and well visualized (Figures 3, S4).

Experiment I – Sparse Forest

This baseline scenario, involving a moving car in a lightly occluded area, is used to demonstrate the tracking precision of the anomaly-based system and to contrast its performance with a conventional object classifier. Anomaly detection is handled by the Reed-Xiaoli (RX) algorithm, which outputs binary anomaly masks later fused into a synthetic aperture anomaly integral used by the multi-UAV platform. The method achieved a high tracking accuracy with an average localization error of 0.26 meters and consistently high anomaly confidence. Figure 4, Table S1, and Figure S2 together provide a good multi-angle evaluation.

YOLO-World v2 is introduced, seemingly as a benchmark, although its role is not explicitly stated. It is not used in decision-making or collective behavior, and no fine-tuning for aerial perspectives is mentioned. The inclusion of only 'vehicle' and 'person' classes is not well justified, particularly since no humans appear in this scenario. Several rows in Table S1 are marked as correct detections despite the absence of YOLO scores, indicating that the anomaly-based method succeeds even when the classifier fails to respond. These omissions suggest a missed opportunity to clarify how YOLO is intended to frame or support the method's evaluation. A benchmarking approach based on YOLO should be clearly defined, starting from the goals of the comparison and identifying a way to ensure fairness of the benchmarking exercise.

Experiment II – Guided Search

This scenario introduces thermal imaging and a human-guided leader drone to assist the multidrone system in localizing two stationary people under dense canopy. Detection errors are low (0.33 m and 0.08 m), and anomaly fusion proves effective even for the prone subject.

The paper explains that the leader drone is manually flown to guide the multi drone fleet into a forested area "presumably containing targets," after which the fleet begins autonomous exploration and PSO-based anomaly detection. However, the paper lacks detail on how guidance is performed (e.g., what the operator sees, how its termination triggers the autonomous phase). Without such detail, it is unclear how operator actions influence fleet initialization or coverage.

To meaningfully evaluate the effectiveness of guided search, metrics such as duration of the guided phase, spatial extent covered during guidance, and the time or number of iterations

required for the system to converge post-guidance would be essential. Similarly, reporting confidence score evolution during and after the transition would help quantify how much the leader drone contributes to accelerating or refining detection. The role of the human confirmation of the detection is not sufficiently justified: what would be the detection performance without human confirmation? Lastly, to truly assess the added value of the method when working in conjunction with human guidance, a benchmark comparison is essential. This could involve evaluating the guided search against scenarios where the human operator explores the area alone, or with only one or two drones under direct control, to determine whether the fleet improves spatial coverage, detection confidence, or response time. Such comparisons would clarify whether the multi drone autonomous capabilities meaningfully enhance human-led search missions, rather than simply following commands without added benefit. Without these benchmarks, it remains difficult to quantify the operational advantage of integrating multi drone autonomy into human-guided search.

Experiment III – Fully Autonomous Tracking

This is the strongest test case: fully autonomous thermal-based tracking of two walking humans in a dense forest. The drone fleet achieves 92.3% precision and recall with 0.53 m average error. Confidence remains high throughout, with only 4 brief dips in those cases when it fails (Figure S3).

However, Table S3 reveals eight failure cases: four where the anomaly detector failed to identify the target, and four where the formation locked onto a false anomaly. These are not discussed in the main text. The lack of explanation for these events—whether due to occlusion, environmental noise, or system limitations—makes it difficult to evaluate the system’s robustness or failure recovery. Additionally, there is no description of how the system responds to false positives or detection absence. Addressing these gaps would strengthen the claims of reliability and field-readiness.

Discussion and Conclusion

The authors reiterate their main contributions and highlight their system’s ability to operate under real-world occlusion and uncertainty. While the conclusion is optimistic, it is grounded in experimental results. We agree with the authors’ positioning of this work as a substantial step forward in field-deployable multi-UAV systems. That said, the conclusion could benefit from a more explicit acknowledgment of system limitations (e.g., centralized computation, no onboard intelligence, false positives, scalability concerns) and more concrete suggestions for future work (e.g., multi-target tracking, online failure recovery, domain adaptation for anomaly detectors).

Summary Evaluation

The revised manuscript addresses several key concerns raised in the initial review, strengthens experimental interpretations and improves transparency in architectural and algorithmic choices. The integration of synthetic aperture imaging with anomaly-based detection is well executed, and the transition from simulation to real-world deployment adds meaningful contributions to the field of aerial sensing with UAV swarms.

However, a few points require minor refinements to ensure the manuscript meets the clarity and reproducibility standards expected for publication.

Remaining Issues and Suggestions

1. Experiment I - Benchmarking against YOLO and legacy method

- The benchmarking against YOLO should be better described. In particular, having access to images from different drones, it should be specified which image was used for object detection: only one, all of them or something in between? It is not specified what is the confidence measure used for YOLO. Also, there is confusion between YOLO confidence and the confidence used for the proposed approach, which are clearly two different things, and should be named in a different way.
- Benchmarking against the legacy method is important to understand the actual contribution, but it would be useful to specify that this is performed in an open loop (much as with YOLO), just using the images taken by the drones during their flight. Results should be added to table S1 as well.
- It is not clear if the reduction in precision with the legacy method can be considered statistically significant or not.

2. Experiment II – Detection of Localized Targets in Dense Forest and Swarm Deployment Evaluation Remains Underdeveloped

- The accompanying video indicates that the two thermal targets (one standing, one lying) are located in separate forest regions, and the 7th “deployment” drone flies to each location sequentially to initialize swarm exploration. This spatial separation is not clearly stated in the manuscript, yet it significantly affects how the swarm detects and tracks anomalies. It appears that at each deployment location, only one target is within the swarm’s sensing footprint, but no details are provided on whether the second target was outside the field of view or ignored by the system. Please clarify.
- It is not clear why the automatic classification is unrealistic here, and why manual confirmation is necessary. In the end, if the threshold $T=2$ is overcome, then a blob gets detected also in this experiment. Why not use this method?
- Why in table S2 are you providing results for a single detection and not a sequence? The swarm was anyway exploring after the deployment drone stopped, hence there should be a sequence of anomaly integrals much as in Experiment I.
- How is it possible that the legacy method did not provide any blob for detection? Given that blobs exist in the anomaly integral, also the legacy method should detect them, although maybe giving more importance to non-target objects. Please provide full data for both the proposed method and the legacy approach.

- Given the stated manual detection confirmation step, how were the errors computed? It would be valuable to quantify the detection effort, including time to detection, swarm path or motion (e.g., center of mass trajectory), and the stopping criterion used to confirm detection.
- 3. Experiment III - target choice and benchmarking against legacy method**
- Why have you chosen two persons walking side by side as a target? Is this because the anomaly of a single person would not be recognisable?
 - It is not clear if the reduction in precision with the legacy method can be considered statistically significant or not.
 - How could you obtain 100% recall for the legacy method (no false negatives) if the proposed approach fails 4 times because of “temporary invisibility (e.g., due to locally dense vegetation)”?
 - Please provide in Table S3 all the results of the legacy method as well.
- 4. Quantitative Boundaries on Occlusion Limits**
- While the discussion section now references prior modeling work ([28]) to support the theoretical scalability of the system under occlusion, the manuscript still lacks concrete operational boundaries that would help assess its real-world robustness (given that each experiment was conducted only once). Specifically, the following factors remain unquantified:
 - i. The maximum level of foliage density at which anomaly detection becomes unreliable.
 - ii. The system’s sensitivity to sensor orientation errors, particularly heading drift beyond the stated $\pm 5^\circ$ tolerance.

These parameters are critical for understanding deployment constraints in real environments. For instance, it is unclear whether the system performs reliably in dense undergrowth, or whether its effectiveness drops sharply beyond a certain occlusion threshold. Including even approximate empirical estimates (e.g., “effective up to 60% canopy coverage at 25 meters altitude”) or describing planned procedures to measure such limits in future work would significantly improve the practical relevance and interpretability of the approach. If such quantification is currently infeasible, a short statement acknowledging this limitation would still enhance clarity.

- How does the system behave in presence of multiple, similarly salient targets? Would the algorithm alternate between them, reject both, or lock onto one arbitrarily? A brief discussion acknowledging these limitations, and their implications for multi-target scenarios, would significantly enhance the transparency and interpretability of this experiment.